# Adaptive Prediction-Powered AutoEval with Reliability and Efficiency Guarantees

**Sangwoo Park   Matteo Zecchin   Osvaldo Simeone**
Department of Engineering
King's College London
London, United Kingdom
{sangwoo.park, matteo.1.zecchin, osvaldo.simeone}@kcl.ac.uk

## Abstract

Selecting artificial intelligence (AI) models, such as large language models (LLMs), from multiple candidates requires accurate performance estimation. This is ideally achieved through empirical evaluations involving abundant real-world data. However, such evaluations are costly and impractical at scale. To address this challenge, autoevaluation methods leverage synthetic data produced by automated evaluators, such as LLMs-as-judges, reducing variance but potentially introducing bias. Recent approaches have employed semi-supervised prediction-powered inference (PPI) to correct for the bias of autoevaluators. However, the use of autoevaluators may lead in practice to a degradation in sample efficiency compared to conventional methods using only real-world data. In this paper, we propose `R-AutoEval+`, a novel framework that provides finite-sample reliability guarantees on the model evaluation, while also ensuring an enhanced (or at least no worse) sample efficiency compared to conventional methods. The key innovation of `R-AutoEval+` is an adaptive construction of the model evaluation variable, which dynamically tunes its reliance on synthetic data, reverting to conventional methods when the autoevaluator is insufficiently accurate. Experiments on the use of LLMs-as-judges for the optimization of quantization settings for the weights of an LLM, for prompt design in LLMs, and for test-time reasoning budget allocation in LLMs confirm the reliability and efficiency of `R-AutoEval+`.

## 1 Introduction

### 1.1 Context and Motivation

Selecting an artificial intelligence (AI) model among multiple candidates necessitates accurately estimating each model's performance. Typically, performance assessment involves actively employing each model to gather relevant empirical evidence or data. To mitigate the substantial cost and practical burden of real-world testing, *autoevaluation* leverages automated tools to evaluate model performance without direct human intervention [35, 31, 12, 8, 52, 5].

Standard evaluation based on human judgment – referred to as `Eval` – and autoevaluation – `AutoEval` – each present distinct advantages and drawbacks. `Eval` provides unbiased estimates of a model's performance but requires costly annotation. In contrast, `AutoEval` is cheaper, as it can rely on abundant synthetic data, but it may introduce estimation bias [35, 12]. Consequently, neither approach, in isolation, ensures *reliable* model evaluation, potentially leading to erroneous evaluation outcomes.

Reliability in evaluation methods can be established via *confidence intervals* that accurately capture the true expected performance at a specified coverage level, or through *testing strategies* that detect whether target performance levels are met at specified false detection probabilities. For `Eval`, both

39th Conference on Neural Information Processing Systems (NeurIPS 2025).

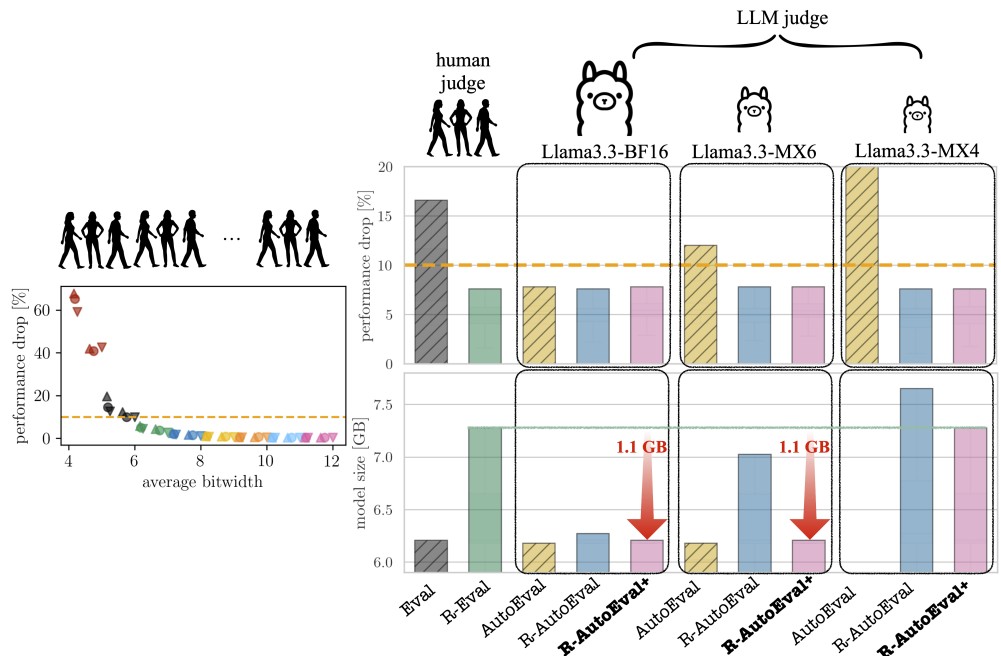

Figure 1: How to select the lightest quantized Llama-3.1-8B-Instruct model [24] (in the MX quantization format [41]) that guarantees up to $10\%$ performance drop as compared to the unquantized version (BF16) (for the TriviaQA task [28])? (left) Ground-truth risk $R$ for different MX quantization settings, requiring massive human-labeled data. (right) Performance drop and corresponding model size for the models chosen via `Eval`, `AutoEval` [35, 31, 52], `R-Eval` [47], `R-AutoEval` [20], and the proposed `R-AutoEval+`. We adopt Llama-3.3-70B-Instruct [24] BF16/MX6/MX4 as the autoevaluators, and set target risk in (1) to $\alpha = 0.1$ and target reliability in (2) to $1 - \delta = 0.9$. Maximum values are reported within the $1.5$ interquartile range (IQR) range [34] across $500$ independent experiments (see Sec. 4 for details).

approaches can be implemented using standard statistical methods [26, 33, 47], which we refer to as `R-Eval`.

Achieving reliability in `AutoEval` is more challenging. Recent works, including [10, 22, 21, 42], have successfully applied a *semi-supervised* inference framework known as *prediction-powered inference* (PPI) [3, 4] to correct for the inherent bias of `AutoEval` by leveraging a small amount of human-labeled, real-world data. These methodologies, referred to here as `R-AutoEval`, either guarantee reliability only *asymptotically* – as the sizes of both synthetic and real-world data sets grow indefinitely [10, 22, 21] – or lack explicit *sample efficiency* guarantees – i.e., they do not provably yield narrower confidence intervals or higher test powers compared to `R-Eval` [20].

In this paper, we introduce `R-AutoEval+`, a novel autoevaluation framework that provides finite-sample (non-asymptotic) reliability guarantees while also ensuring improved (or at least no worse) sample efficiency compared to `R-AutoEval`. The primary innovation of `R-AutoEval+` is its sequential construction of the model evaluation variable, enabling it to *adaptively* adjust its reliance on synthetic data based on evolving assessments of the autoevaluator's quality. `R-AutoEval+` seamlessly reverts to `R-Eval` when synthetic data are deemed to be of insufficient quality, while otherwise employing a weighted variant of `R-AutoEval` to enhance efficiency.

At a technical level, `R-AutoEval+` leverages two primary methodologies. The first is the game-theoretic *testing-by-betting* approach [43, 38] to mean estimation introduced by [47]. The second is `PPI++` [4], an enhanced variant of `PPI` that incorporates a regularization coefficient to control the estimator's dependence on autoevaluator-generated data [4].

## 1.2 Overview of the Main Results

To clarify the main concepts, consider the problem of selecting a large language model (LLM) from multiple candidate models characterized by different sizes. These models are derived from a single base LLM through quantization with varying average bitwidths using the MX format [41]. As illustrated in left panel of Fig. 1, our goal is to identify models that maintain performance comparable to the full-precision baseline, tolerating at most a predefined performance degradation threshold $\alpha$ (e.g., $\alpha = 10\%$ in the figure).

Formally, consider a bounded loss function $\ell(X, Y) \in [0, 1]$ that measures the performance of a given candidate model. In the context of Fig. 1, this loss quantifies the performance gap between the candidate quantized LLM and its full-precision counterpart. Our primary objective is to verify whether or not the expected loss, or *risk*, defined as $R = \mathbb{E}[\ell(X, Y)]$, exceeds a target level $\alpha$, i.e., whether the following *risk-controlling condition* can be satisfied

$$R \leq \alpha. \tag{1}$$

The challenge is that evaluating the risk $R$ necessitates access to a large number of *human-generated*, i.e., *real-world*, responses $Y$ corresponding to queries $X$.

### 1.2.1 `Eval` and `AutoEval`

Given $n$ pairs of real-world data $\mathcal{D}_n = \{(X_i, Y_i)\}_{i=1}^n$ with $(X_i, Y_i) \overset{\text{i.i.d.}}{\sim} P_{XY} = P_X P_{Y|X}$ for $i = 1, ..., n$, the standard `Eval` approach estimates the risk $R$ for any given candidate model via the empirical average $\hat{R}_n = (1/n) \sum_{i=1}^n \ell(X_i, Y_i)$.

In contrast, `AutoEval` does not assume access to real data $\mathcal{D}_n$, relying instead the availability of a pre-trained *autoevaluator* $f : \mathcal{X} \to \mathcal{Y}$, where $\mathcal{X}$ and $\mathcal{Y}$ denote the respective domains of input and output $X, Y$. In the example of Fig. 1, the autoevaluator is a larger LLM, serving as an *LLM judge* [52]. Specifically, given an *unlabeled* data $\mathcal{D}_N^{\text{unl}} = \{\tilde{X}_i\}_{i=1}^N$ with $\tilde{X}_i \overset{\text{i.i.d.}}{\sim} P_X$, `AutoEval` estimates the risk as $\hat{R}_N^f = (1/N) \sum_{i=1}^N \ell(\tilde{X}_i, f(\tilde{X}_i))$, thus using the outputs from the LLM judge as labels [35, 31, 52].

Lacking formal uncertainty quantification, the risk-controlling condition (1) cannot be guaranteed via the risk estimates $\hat{R}_n$ and $\hat{R}_N^f$ provided by `Eval` and `AutoEval`. That is, even if the evaluation outcome for a candidate model satisfies the inequality $\hat{R}_n \leq \alpha$, or $\hat{R}_N^f \leq \alpha$, its actual performance may still fail to meet the condition (1). In the example shown in Fig.1, LLMs selected via `Eval` and `AutoEval` exhibit performance degradations that exceed the target threshold of $\alpha = 10\%$. Furthermore, for `AutoEval`, the performance degradation becomes larger as the quality of the LLM judge deteriorates.

### 1.2.2 `R-Eval` and `R-AutoEval`

*Reliable Eval* (R-Eval) [47] and *Reliable AutoEval* (R-AutoEval) [20] endow `Eval` and `AutoEval`, respectively, with reliability guarantees by formulating model evaluation as a binary hypothesis test problem. Accordingly, these evaluation protocols test the null hypothesis $\mathcal{H}_0 : R > \alpha$ that the model's risk exceeds the target $\alpha$ against the alternative hypothesis $\mathcal{H}_1 : R \leq \alpha$ that the risk-controlling condition (1) is satisfied.

Define as $T_n \in \{0, 1\}$ the test output, with $T_n = 1$ indicating the decision for the alternative hypothesis $\mathcal{H}_1 : R \leq \alpha$. Then, an evaluation procedure is said to be *reliable* at level $1 - \delta$ if the probability of incorrectly concluding that a candidate model satisfies the requirement (1) does not exceed the level $\delta \in (0, 1)$, i.e.,

$$\Pr\left[T_n = 1 | R > \alpha\right] \leq \delta. \tag{2}$$

While `Eval` constructs the test decision based only on real data $\mathcal{D}_n$, `R-AutoEval` incorporates also synthetic data by leveraging PPI [3] as we detail in Sec. 2.

As illustrated in Fig. 1, selecting the smallest LLM among the candidates deemed risk-controlling by `R-Eval` and `R-AutoEval`, i.e., the models with respective testing results being $T_n = 1$, indeed results in a performance drop that remains within the tolerated risk level of $\alpha = 10\%$. However, as also shown in Fig. 1, `R-AutoEval` may select a model with a larger size than `R-Eval`, highlighting the potential inefficiency due to the use of an LLM judge.

### 1.2.3 `R-AutoEval+`

This paper proposes `R-AutoEval+`, a novel reliable autoevaluation method that adaptively tunes its reliance on synthetic data based on an evolving reliability assessments of the autoevaluator. This approach balances synthetic and real data, reverting to conventional methods when the accuracy of the autoevaluator is insufficient, while maintaining rigorous statistical guarantees.

`R-AutoEval+` is not only reliable in the sense of satisfying the condition (2), but it also provides a guaranteed improvement (possibly not strict) in terms of sample efficiency. To formalize this property, define the *sample complexity* of an evaluation method producing test variable $T_n$ as the smallest average size of the real-world data set $\mathcal{D}_n$ necessary to conclude that a model satisfies the requirement (1) under the reliability constraint (2), i.e., [48]

$$n_{\min}(\delta) = \mathbb{E}[\min\{n : T_n = 1\}|R \leq \alpha]. \tag{3}$$

An evaluation scheme with smaller sample complexity $n_{\min}(\delta)$ can generally identify more efficient candidate models with the same amount of real data. The main result is informally outlined as follows.

**Theorem 1** (Informal). *Under mild regularity assumptions, for sufficiently low tolerated unreliability level $\delta$, `R-AutoEval+` is provably more sample efficient than both `R-Eval` [47] and `R-AutoEval` [20], i.e.,*

$$n_{min}^{\texttt{R-AutoEval+}}(\delta) \leq \min\left\{n_{min}^{\texttt{R-Eval}}(\delta), n_{min}^{\texttt{R-AutoEval}}(\delta)\right\}. \tag{4}$$

*Furthermore, this inequality is strict when the autoevaluator is sufficiently accurate.*

## 2 Preliminaries: `R-Eval` and `R-AutoEval`

In this section, we first review the testing-by-betting framework [43, 37, 47, 48, 38], and then review `R-Eval` [47] and `R-AutoEval` [20]. Further connections to the state-of-the-art can be found in Appendix A.

### 2.1 Testing-By-Betting

Consider the problem of testing the null hypothesis $\mathcal{H}_0 : R > \alpha$ against the alternative hypothesis $\mathcal{H}_1 : R \leq \alpha$ based on the observations of $n$ i.i.d. bounded random variables $\{q_i\}_{i=1}^n$ with $q_i \in [m, M]$ providing unbiased estimates of the risk, i.e., $\mathbb{E}[q_i] = R$. An e-value $E_n$ is a nonnegative statistic of the observations $\{q_i\}_{i=1}^n$ whose expectation under the null hypothesis does not exceed 1, i.e., $\mathbb{E}[E_n|R > \alpha] \leq 1$ [45]. An e-value $E_n$ can be interpreted as providing evidence *in favor* of the alternative hypothesis $\mathcal{H}_1$, i.e., of the risk-controlling condition (1) [43]. With an e-value $E_n$, the test

$$T_n = \mathbb{1}(E_n \geq 1/\delta) \tag{5}$$

meets the reliability condition (2) due to Markov's inequality for a fixed $n$.

The testing-by-betting approach constructs an e-value $E_n$ sequentially by processing the observations $q_i$ one by one over index $i = 1, ..., n$. Specifically, an e-value can be obtained via the product of the contributions of each observation $q_i$ as [47, 6]

$$E_n = \prod_{i=1}^n \left(1 - \lambda_i(q_i - \alpha)\right), \tag{6}$$

where $E_0 = 1$ and $\lambda_i \in [0, 1/(M - \alpha))$ is an arbitrary function of the past observations $\{q_j\}_{j=1}^{i-1}$. To verify that the quantify (6) is an e-value, one can use the independence of the observations $\{q_i\}_{i=1}^n$ and the unbiasedness assumption $\mathbb{E}[q_i] = R$ as $\mathbb{E}[E_n|R > \alpha] = \prod_{i=1}^n (1 - \lambda_i(\mathbb{E}[q_i|R > \alpha] - \alpha)) \leq 1$.

The e-value has an interpretation in terms of a sequential betting game, which supports the design of the sequence of bets $\lambda_i$ using online convex optimization [47]. In this game, at each round $i$, based on the observations $\{q_j\}_{j=1}^{i-1}$, a gambler bets an amount of her wealth measured by variable $\lambda_i$ on the outcome $q_i \leq \alpha$ that the next observation $q_i$ does not exceed the target $\alpha$. Accordingly, the e-value $E_n$ in (6) represents the wealth accumulated after $n$ rounds of betting with initial wealth $E_0 = 1$ [47].

A more general form of e-value has been also proposed that allows the gambler to have $S \geq 1$ different betting strategies $\{\lambda_{s,i}\}_{s=1}^{S}$, with each strategy $\lambda_{s,i}$ being responsible for a fraction $w_{s,i}$ of the wealth. To elaborate, fix a probability vector $w_i = [w_{1,i}, ..., w_{S,i}]$, which, like the betting strategies $\{\lambda_{s,i}\}_{s=1}^{S}$, may depend on the past observations $\{q_j\}_{j=1}^{i-1}$. The resulting e-value is defined as the convex combination [47, B.8]

$$E_n = \prod_{i=1}^{n} \sum_{s=1}^{S} w_{s,i}\big(1 - \lambda_{s,i}(q_i - \alpha)\big). \tag{7}$$

## 2.2 `R-Eval` and `R-AutoEval`

Both `R-Eval` and `R-AutoEval` follow the testing-by-betting approach presented in the previous subsection. Specifically, `R-Eval` computes the e-value $E_n^{\texttt{R-Eval}}$ in (6) using directly the observations $q_i = \ell_i$ of the losses $\ell_i = \ell(X_i, Y_i) \in [m = 0, M = 1]$, obtaining the decision $T_n^{\texttt{R-Eval}}$ in (5).

`R-AutoEval` [20] also computes the e-value $E_n^{\texttt{R-AutoEval}}$ in (6), but with effective observations $q_i = \ell_i^f$ that incorporate both real data and synthetic data. In particular, each effective observation $q_i = \ell_i^f$ uses the real data point $(X_i, Y_i) \in \mathcal{D}_n$ together with $r = \lfloor N/n \rfloor$ autoevaluated samples $\{(\tilde{X}_i, f(\tilde{X}_i))\}_{r \cdot (i-1)+1}^{r \cdot i}$ with $\tilde{X}_i \in \tilde{\mathcal{D}}_N$. Note that this divides the set of unlabeled samples $\tilde{\mathcal{D}}_N$ equally across the effective observations.

Specifically, `R-AutoEval` uses effective observations obtained via PPI [3] by correcting the bias of the empirical autoevaluated risk using the real data sample as

$$\ell_i^f = \underbrace{\frac{1}{r} \sum_{i'=r \cdot (i-1)+1}^{r \cdot i} \ell(\tilde{X}_{i'}, f(\tilde{X}_{i'}))}_{\text{autoevaluator data}} + \underbrace{\ell(X_i, Y_i) - \ell(X_i, f(X_i))}_{\text{bias correction}}. \tag{8}$$

One can readily check that this is an unbiased estimate of the risk, i.e., $\mathbb{E}[\ell_i^f] = R$, with support interval $\ell_i^f \in [m = -1, M = 2]$. However, owing to the possible lower quality of the autoevaluator's labels $f(\tilde{X}_i)$, using the effective observations (8) may result in a larger sample complexity compared to `R-Eval` (see Fig. 1).

# 3 `R-AutoEval+`: AutoEval with Reliability and Efficiency Guarantees

In this section, we present the proposed autoevaluation method, `R-AutoEval+`. `R-AutoEval+` aims at balancing the importance assigned to synthetic and real-world data in the effective observations (8). Through the proposed mechanism, `R-AutoEval+` reverts to using the effective observations $q_i = \ell_i$ of `R-Eval` when the autoevaluator is inaccurate, while leveraging the effective observations of `R-AutoEval`, $q_i = \ell_i^f$, when the autoevaluator is very accurate. The approach is based on an adaptive construction of the effective observations used in the e-value statistic (7), dynamically tuning the reliance on synthetic data based on evolving reliability assessments of the autoevaluator.

The key idea is to weight the contribution of synthetic data in the e-value with a factor $\rho \in [0, 1]$, so that setting $\rho = 0$ recovers `AutoEval`, while setting $\rho = 1$ yields `R-AutoEval`. In order to automatically identify the best value of the factor $\rho$, `R-AutoEval+` processes the real-world data samples sequentially across index $i = 1, ..., n$, tracking the performance of $S$ possible candidate values $\{\rho_s\}_{s=1}^{S}$, with

$$0 = \rho_1 < \cdots < \rho_S = 1. \tag{9}$$

Specifically, `R-AutoEval+` maintains adaptive weights $\{w_{s,i}\}_{s=1}^{S}$ across index $i = 1, ..., n$, with weight $w_{s,i}$ associated to candidate value $\rho_s$. The resulting effective observations are used in (7).

After providing a more detailed description of `R-AutoEval+`, this section demonstrates that `R-AutoEval+` can provably enhance the sample efficiency of both `R-Eval` and `R-AutoEval`.

## 3.1 Adaptive Effective Observations and E-Values

At each round $i$, `R-AutoEval+` computes $S$ effective observations $\{\ell^f_{s,i}\}^S_{s=1}$, with each effective observation $\ell^f_{s,i}$ weighting the contribution of synthetic data via the factor $\rho_s$. In particular, generalizing (8) via `PPI++` [4], each $s$-th effective observation is given by

$$\ell^f_{s,i} = \underbrace{\frac{\rho_s}{r} \sum_{i'=r\cdot(i-1)+1}^{r\cdot i} \ell(\tilde{X}_{i'}, f(\tilde{X}_{i'}))}_{\text{autoevaluator data}} + \underbrace{\ell(X_i, Y_i) - \rho_s \cdot \ell(X_i, f(X_i))}_{\text{bias correction}}, \tag{10}$$

where the factor $\rho_s$ multiplies the contributions of the autoevaluator. The observation (10) is a bounded unbiased estimate of the risk $R$, i.e., $\mathbb{E}[\ell^f_{s,i}] = R$, with $\ell^f_{s,i} \in [m = -\rho_s, M = 1 + \rho_s]$ [4].

Using the effective observations $\{q_{s,i} = \ell^f_{s,i}\}^S_{s=1}$, `R-AutoEval+` constructs a variant of the e-value $E_n$ in (7) in which a different bounded unbiased estimate of the risk, namely $\ell^f_{s,i}$, is used for each $s$-th term, i.e.,

$$E_n = \prod_{i=1}^{n} \sum_{s=1}^{S} w_{s,i}\big(1 - \lambda_{s,i}(\ell^f_{s,i} - \alpha)\big). \tag{11}$$

Note that this is indeed a valid e-value, i.e., $\mathbb{E}[E_n | R > \alpha] \leq 1$.

In the e-value (11), the weights $w_{s,i}$ associated with each factor $\rho_s$ are updated over index $i = 1, ..., n$ depending on the evidence accumulated up to round $i - 1$. To do this, define as

$$E_{s,i} = \prod_{j=1}^{i} \big(1 - \lambda_{s,j}(\ell^f_{s,j} - \alpha)\big) \tag{12}$$

the e-value (6) computed using only the effective observations $\ell^f_{s,i}$ up to round $i$. Intuitively, a larger value of the quantity $E_{s,i}$ indicates that the factor $\rho_s$ yields a large evidence in favor of the alternative (risk-controlling) hypothesis (1).

Following this logic, the weight $w_{s,i}$ is updated as

$$w_{s,i} = \frac{w_{s,0} \cdot E_{s,i-1}}{\sum_{s'=1}^{S} w_{s',0} \cdot E_{s',i-1}}, \tag{13}$$

for all $s = 1, ..., S$, with initial strictly positive weights $\{w_{s,0}\}^S_{s=1}$, where $\sum_{s=1}^{S} w_{s,0} = 1$ and $E_{s,0} = 1$. By (13), values of the factor $\rho_s$ associated with a larger evidence $E_{s,i-1}$ in favor of the risk-controlling condition (1) are assigned a proportionally larger weight in the e-value (7).

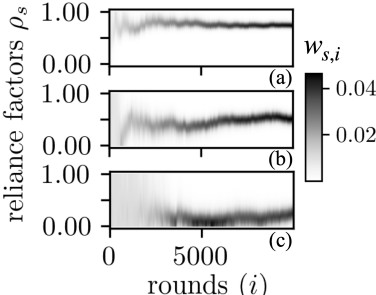

Figure 2: Heatmap of the evolution of the weights $\{w_{s,i}\}^{100}_{s=1}$ assigned to the factors $\rho_s$ as a function of the processing round $i$ for Example 1. The autoevaluator reports the correct loss with probability $\gamma = 0.99$ (top), $\gamma = 0.9$ (middle), and $\gamma = 0.7$ (bottom). `R-AutoEval+` assigns larger weights to synthetic data, i.e., to larger values of $\rho_s$, when the autoevaluator is of higher quality.

The update (13) aims at identifying the most informative effective observations $\ell^f_{s,i}$ for $s = 1, ..., S$ by sequentially processing the available data. The following simple example demonstrates its operation.

***Example* 1.** *Consider a setting where the autoevaluator yields the same output as the human judge with probability $\gamma$. Specifically, assume a binary loss $\ell(X, Y) \in \{0, 1\}$ with mean equal to the risk $R = 0.1$, and on autoevaluated loss $\ell(X, f(X))$ given by $\ell(X, f(X)) = \ell(X, Y) \cdot (1 - \epsilon) + (1 - \ell(X, Y)) \cdot \epsilon$, where $\epsilon \in \{0, 1\}$ has mean $1 - \gamma$. Fix the target risk $\alpha = 0.12$ and the synthetic-to-real data ratio $r = 10$.*

*Fig. 2 provides an heatmap of the evolution of the weights $\{w_{s,i}\}^S_{s=1}$ over index $i$ for $S = 100$ uniformly spaced candidate factors in the range $[0, 1]$. We set initial weights $w_{s,0} = 1/S$ for $s = 1, ..., S$. The top figure corresponds to $\gamma = 0.99$, the middle figure to $\gamma = 0.9$ and the bottom figure to $\gamma = 0.7$, so that the quality of the autoevaluator decreases in going from the top panel to the bottom panel. The figure shows that, as the autoevaluator becomes less reliable, the update (13) correctly decreases the reliance of the e-value (11) on the effective observations that leverage synthetic data. Specifically, it is seen from the figure that the weights $\{w_{s,i}\}^S_{s=1}$ tend to concentrate around the values 0.9, 0.5, and 0 for $\gamma = 0.99, 0.9,$ and $0.7$, respectively.* □

The overall procedure of `R-AutoEval+` is summarized in Algorithm 1 (Appendix B).

## 3.2 Sample Efficiency Guarantees

In this section, we analyze the sample efficiency of `R-AutoEval+`. We start by reviewing existing results on `R-Eval` and `R-AutoEval`, and then we introduce a formal version of Theorem 1 (see Sec. 1) on the sample efficiency of `R-AutoEval+`.

### 3.2.1 Sample complexity of `R-Eval` and `R-AutoEval`

Reference [48] showed that the sample complexity of the e-value (6) can be analyzed by considering its maximum expected logarithmic per-round increment under the alternative hypothesis, i.e.,

$$g_\star = \mathbb{E}[\log(1 - \lambda_\star(q_i - \alpha))|R \leq \alpha], \tag{14}$$

where $\lambda_\star$ is the optimal constant betting variable $\lambda_\star = \arg\max_{\lambda \in [0,1/(M-\alpha))} \mathbb{E}[\log(1 - \lambda(q_i - \alpha))|R \leq \alpha]$. Specifically, defining $E_n(\lambda_\star) = \prod_{i=1}^n (1 - \lambda_\star(q_i - \alpha))$ for the e-value (6) with constant $\lambda_i = \lambda_\star$ for all rounds $i = 1, ..., n$, we have the following result.

**Theorem 2** (Sample complexity of testing-by-betting (6) [48, Theorem 3.3]). *Assume that: (A1) the betting strategy $\lambda_i$ admits sublinear regret with respect to the optimal constant $\lambda_\star$, i.e., $\log E_n(\lambda_\star) - \log E_n = o(n)$ (see Appendix C, for an example of such betting strategy), and (A2) the instantaneous logarithmic increment of the e-value $E_n(\lambda_\star)$ has finite $\sigma$-th central moment for some $\sigma > 2$, i.e., $\mathbb{E}[|\log(1 - \lambda_\star \cdot (q_i - \alpha)) - g_\star|^\sigma|R \leq \alpha] < \infty$. Then, the sample complexity (3) obtained by the test variable $T_n$ in (5) with the e-value $E_n$ in (6) admits the limit*

$$\lim_{\delta \to 0^+} \frac{n_{min}(\delta)}{\log(1/\delta)} = \frac{1}{g_\star}. \tag{15}$$

To interpret this result, consider the second-order Taylor approximation $\log(1 - y) \simeq -y - y^2/2$, which yields (see also [48, Sec. 4.2])

$$\frac{1}{g_\star} \simeq 2\left(1 + \frac{\text{Var}(q_i|R \leq \alpha)}{(\alpha - R)^2}\right) \tag{16}$$

According to this approximation, the sample complexity grows with the variance $\text{Var}(q_i|R \leq \alpha) = \mathbb{E}[|q_i - R|^2|R \leq \alpha]$ of the observation under the alternative hypothesis.

The result (15) can be readily applied to obtain the sample complexities of `R-Eval` and `R-AutoEval` by setting $q_i = \ell_i$ and $q_i = \ell_i^f$, respectively, in the e-value (6). In the presence of a low-quality autoevaluator, it is known that `PPI` can increase the variance of the risk estimates [3, 4, 22, 10]. Therefore, based on the approximation (16), `R-AutoEval` may have a larger sample complexity than `R-Eval` when the autoevaluation is not sufficiently accurate.

### 3.2.2 Sample complexity of `R-AutoEval+`

`R-AutoEval+` optimizes not only the $S$ betting strategies $\{\lambda_{s,i}\}_{s=1}^S$, but also the weights $\{w_{s,i}\}_{s=1}^S$ associated to the factors $\{\rho_s\}_{s=1}^S$ determining the reliance of the effective observations (10) on the autoevaluated data. Since the weight update (13) has the form of *exponential weights* [23], the following property follows from existing results on online convex optimization [18, Lemma 1].

**Lemma 1** (Sublinear regret for the weights (13)). *For any set of betting strategies $\{\lambda_{s,i}\}_{s=1}^S$ and for any positive initial weights $\{w_{s,0}\}_{s=1}^S$, the weight update strategy (13) satisfies*

$$\max_{s=1,...,S} \underbrace{\sum_{i=1}^n \log\left(1 - \lambda_{s,i}(\ell_{s,i}^f - \alpha)\right)}_{\log E_{s,n}} - \underbrace{\sum_{i=1}^n \log \sum_{s=1}^S w_{s,i}\left(1 - \lambda_{s,i}(\ell_{s,i}^f - \alpha)\right)}_{\log E_n} \leq \max_{s=1,...,S} \log\left(\frac{1}{w_{s,0}}\right). \tag{17}$$

Intuitively, this result implies that the weight update (13) can identify the best factor $\rho_s$ in the set $\{\rho_s\}_{s=1}^S$, determining an optimal level of reliance on autoevaluated data. In fact, the first term in

(17) represents the maximum, i.e., most informative, e-value among all the e-values $\{E_{s,n}\}_{s=1}^{S}$ corresponding to the $S$ candidate factors $\{\rho_s\}_{s=1}^{S}$. This result, in turn, suggests that R-AutoEval+ may be able to outperform both R-Eval and R-AutoEval, reducing to the former when autoevaluated data is of poor quality and to the latter when the autoevaluated data is sufficiently accurate. The following result formalizes this intuition.

**Theorem 3** (Sample complexity of R-AutoEval+). *For every $s = 1, ..., S$, suppose that the betting strategy $\lambda_{s,i}$ satisfies assumptions (A1) and (A2) in Theorem 2 with $E_{s,n}$ in lieu of $E_n$ and $q_i = \ell_{s,i}^{f}$. Then, the sample complexity of R-AutoEval+ satisfies the following limit*

$$\lim_{\delta \to 0^+} \frac{n_{min}^{\texttt{R-AutoEval+}}(\delta)}{\log(1/\delta)} \leq \min_{s=1,...,S} \left\{ \frac{1}{g_{s,\star}} \right\}. \tag{18}$$

*Proof.* See Appendix F.2. □

The limit (18) implies the main result (4) in Theorem 1. In fact, by (15), the ratio $1/g_{s,\star}$ in (18) corresponds to the scaling of the sample complexity of R-Eval and R-AutoEval by setting $s = 1$ and $s = S$, respectively. The next example shows that the inequality in (4) can be strict.

**Example 2.** *Consider again the example setting in Example 1. The sample complexity $n_{min}(\delta)$ is plotted in the top part of Fig. 3 as a function of $\log(1/\delta)$ for $\alpha = 0.12$, $S = 10$, $r = 10$, and for (a) $\gamma = 0.99$, (b) $\gamma = 0.9$, and (c) $\gamma = 0.7$. The results are averaged over $100$ independent experiments and we use the universal portfolio betting strategy (see Appendix C) [15, 16, 48]. The figure confirms the linear trend of the sample complexity with respect to the term $\log(1/\delta)$. Furthermore, it shows that the inequality (4) can indeed be strict, with R-AutoEval+ outperforming both R-Eval and R-AutoEval.*

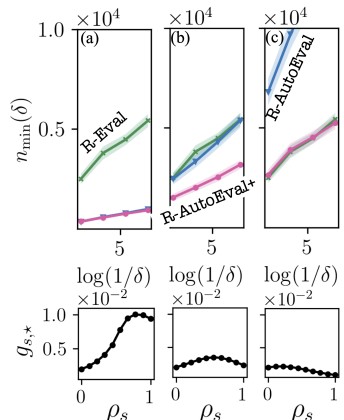

*The bottom part of the figure plots the maximum expected logarithmic increment of the e-value, $g_{s,\star}$, in (18) as a function of the factor $\rho_s$ for (a) $\gamma = 0.99$, (b) $\gamma = 0.9$, (c) $\gamma = 0.7$. As the autoevaluator becomes less (more) reliable, i.e., as $\gamma$ decreases (increases), the maximum value of the expected increment $g_{s,\star}$ is obtained for values of $\rho_s$ closer to zero (one), making R-AutoEval+ behave as R-Eval (R-AutoEval).*

Figure 3: Sample complexity as a function of $\log(1/\delta)$ (top) and maximum expected increment of the log-e-value $g_{s,\star}$ as a function of $\rho_s$ (bottom) for Example 2.

More generally, using the approximation (16), one can conclude that the sample complexity of R-AutoEval+ is strictly smaller than for R-Eval and R-AutoEval as long as the variance of the effective observation $\ell_{s,i}^{f}$ in (10) for some index $s$ different from $s = 1$ and $s = S$ is strictly smaller than for the effective observations $\ell_i$ and $\ell_i^{f}$ in (8). As shown in [4, Example 6.1], this condition is satisfied when the autoevaluator is sufficiently accurate.

## 4 Experimental Results[1]

For experimental validation, we consider three model selection applications: 1) selecting the lightest quantized LLM with guaranteed performance drop as compared to the baseline model on the TriviaQA data set [28] (see Fig. 1); 2) selecting the shortest prompt template for an LLM with guaranteed accuracy on the Instruct-Induction task [27]; and 3) test-time reasoning budget allocation with guaranteed performance enhancement on the GSM8K data set [13]. We set $S = 10$ with $\rho_s$ being uniformly spaced in the range $[0, 1]$ and choose initial weights as $w_{s,0} = 1/S$. We refer to Appendix E for results with different choices of such hyperparameters. All the results in this section are reported after averaging over 100 independent experiments, and 2 H100 GPUs are used for LLM executions.

**1) Selecting quantized LLMs:** As shown in Fig. 1, for this task the candidate set of models consists of LLMs obtained from the baseline Llama-3.1-8B-Instruct by applying quantization in

---

[1]Code is available at `https://github.com/kclip/R_AutoEval_plus`.

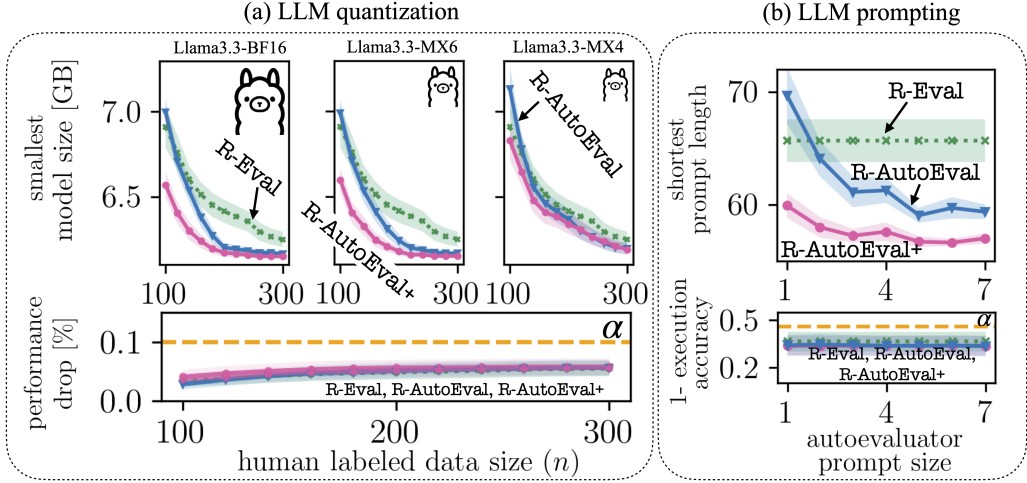

Figure 4: Risk-controlling model selection using `R-Eval` [47, 6, 2], `R-AutoEval` [20], and the proposed `R-AutoEval+` for the problems of (a) selecting the lightest quantized LLM with guaranteed performance drop on the TriviaQA data set [28], and (b) selecting the shortest prompt template with guaranteed execution accuracy on the Instruction-Induction task [27]. (a, top) Size of the smallest selected model versus the number $n$ of real-world data points, and (a, bottom) corresponding performance drop. (b, top) Length of the shortest selected prompt template versus the number of in-context samples used by the autoevaluator, and (b, bottom) corresponding complement of the execution accuracy.

the MX format [40]. The candidate formats have different configurations specified by parameters $[k_1, k_2, d_1, d_2, \mathcal{M}] \in \mathbb{N}^5$ where $k_1, k_2$ are the first, second block granularity levels, $d_1, d_2$ are the first, second scale bit-width levels, and $\mathcal{M}$ is the mantissa bit-width [40, Table II]. We set $k_1 \in \{16, 64\}, d_1 = 8, d_2 = 1, \mathcal{M} \in \{3, 4, ..., 10\}$ with $k_1/k_2 = \{2, 4, 8\}$. Model selection is carried out based on fixed sequence testing (FST) [7], visiting the candidates in order of decreasing average bitwidth. FST guarantees the family-wise error rate (FWER) with error probability no larger than $\delta$.

For the autoevaluator, we adopt a larger Llama-3.1-70B-Instruct [24], whose quality is controlled by adjusting the weight precision of the autoevaluator from full-precision BF16 to MX6/MX4, with average bitwidth decreasing from 16 to 6/4 [41]. We set $\delta = 0.1, \alpha = 0.1, n = 150$, and $r = 5$ for Fig. 1 while vary $n$ from 100 to 300 with $r = 3$ for Fig. 4. Fig. 1 reports maximum values within the 1.5 interquartile range (IQR) range [34] across 500 independent experiments (the corresponding full box plot can be found in Appendix E).

Fig. 4(a) plots the size of the smallest selected model versus the number $n$ of real-world data points (top) and the corresponding performance drop against the full-precision model (bottom). The figure confirms that all schemes do not exceed the target maximum performance drop $\alpha = 0.1$. Furthermore, `R-AutoEval+` outperforms both `R-Eval` and `R-AutoEval` when the autoevaluator is sufficiently accurate (Llama3.3-BF16/MX6), while recovering `R-Eval` when the autoevaluator is of low quality (Llama3.3-MX4), thus confirming the efficiency guarantee of `R-AutoEval+` in Theorem 3.

**2) Selecting the shortest prompt template:** For this second task, the candidate set consists of 25 prompt templates designed to enhance the zero-shot performance of Llama-3.1-8B-Instruct using the larger Llama-3.1-70B-Instruct [24] via the forward mode generation of automatic prompt engineering (APE) [53]. Model selection is carried out via the Bonferroni correction [9], thus applying the test (5) with $\delta/25$ in lieu of $\delta$.

For the autoevaluator, we adopt in-context learning [11] on the Llama-3.1-70B-Instruct with prompt examples randomly chosen from the same held-out data set used for APE. The accuracy of the autoevaluator is controlled by varying the number of prompt examples from 1 to 7. We set $\delta = 0.1, n = 200, r = 9$, with $\alpha$ chosen as the minimum value in the set $\{0.05, 0.1, ..., 0.95\}$ for which `R-AutoEval` [20] finds at least one reliable prompt template with the strongest autoevaluator. We select the longest prompt template if the model selection algorithm does not select any template.

Table 1: Selecting the smallest reasoning budget for Qwen3-1.7B that ensures at least 3% accuracy enhancement as compared to its non-reasoning mode, evaluated on GSM8K data set [13]: average number of generated tokens with standard deviation shown within parentheses.

| autoevaluator (accuracy) | R-Eval | R-AutoEval | R-AutoEval+ |
|---|---|---|---|
| BitNet b1.58 (35%) | | 950.27 (152.84) | **942.47 (135.65)** |
| Llama-3.2-3B-Instruct (66%) | | 900.58 (122.70) | **892.86 (112.10)** |
| Qwen3-32B (82%) | 983.34 (137.87) | 1007.45 (150.48) | **941.20 (129.61)** |
| DeepSeek-R1-Distill-Qwen-32B (89%) | | 893.39 (105.13) | **866.22 (80.58)** |
| Llama-3.3-70B-Instruct (89%) | | 854.42 (90.26) | **847.05 (69.21)** |
| GPT-4.1 (93%) | | 883.99 (103.00) | **856.13 (70.93)** |

Fig. 4(b) plots the length of the shortest selected prompt template versus the number of in-context samples used for autoevaluator (top) and the corresponding risk defined as the complement of the execution accuracy as defined in [27, A.1]. The figure confirms that all schemes reach execution accuracy no smaller than $1 - \alpha$. Moreover, R-AutoEval+ consistently outperforms both R-Eval and R-AutoEval irrespective of the number of in-context samples used for the autoevaluator.

**3) Test-time reasoning budget allocation:** For the last task, the candidate set consists of computation budgets for the reasoning mode of the Qwen3-1.7B [51] base model, varying between 128 and 1280 tokens. Model selection is carried out via FST, visiting the candidates in order of decreasing reasoning budget. For the autoevaluator, we adopt different kinds of pre-trained LLMs, ranging from large-scale models such as GPT-4.1 [1] to light-weight models such as BitNet b1.58 [32]. We set $\delta = 0.1, n = 1000, r = 4$, and $\alpha = 0.03$ (i.e., reasoning should improve accuracy by at least 3%).

Table 1 confirms again the efficiency gain of R-AutoEval+ as compared to R-Eval and R-AutoEval, saving up to 127 tokens over R-Eval and up to 66 tokens over R-AutoEval on average. Choosing the autoevaluator from the same family of the model is seen to substantially reduce the gain of autoeval-based approaches. For instance, Llama-3.2-3B-Instruct autoevaluator achieves much lower accuracy than Qwen3-32B autoevaluator (66% vs. 82%) but it significantly helps reducing the number of reasoning tokens: R-AutoEval+ saves 90 tokens over R-Eval when using Llama-3.2-3B-Instruct autoevaluator, while it saves 42 tokens over R-Eval when using Qwen3-32B autoevaluator; R-AutoEval saves 83 tokens over R-Eval when using Llama-3.2-3B-Instruct autoevaluator, while it requires 24 tokens more than R-Eval when using Qwen3-32B autoevaluator. Such behavior can be understood as a consequence of positive feedback of LLM judges within the same family [52], also known as preference leakage [30], which makes the bias correction (8) more challenging.

We refer to Appendix E for further details and additional experiments.

## 5   Conclusion and Further Discussions

This work introduced R-AutoEval+, a novel autoevaluation method that can provably enhance the efficiency of the state-of-the-art evaluation method while maintaining strict finite-sample reliability guarantees. The theoretical properties of R-AutoEval+ were confirmed by experimental results on LLM quantization and LLM prompting with LLM judges as autoevaluators.

Some limitations of this work are as follows: (*i*) R-AutoEval+ requires access to real-world unlabeled data; (*ii*) the discrete set of candidate factors determining reliance on synthetic data are fixed *a priori*; and lastly, (*iii*) the sample efficiency guarantee in Theorem 3 only holds for sufficiently high target reliability levels $1 - \delta$. Addressing these limitations may leverage the tools in [3, 16, 36], and we leave these directions to future work.

Another interesting direction for future research includes combining the benefits of R-AutoEval+ in adaptively weighting synthetic data with the complementary advantages of methods that actively select real data [49, 46, 54, 17, 50].

## Acknowledgments

This work was supported by the European Union's Horizon Europe project CENTRIC (101096379). The work of Osvaldo Simeone was also supported by the Open Fellowships of the EPSRC (EP/W024101/1) and by the EPSRC project (EP/X011852/1). Thanks also to the Advanced Research and Invention Agency (ARIA) for supporting broader foundational work in this space. The authors also thank Bipin Rajendran for his advice on LLM quantization.

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

## A   Further Related Work

**Active evaluation.** While standard `Eval` leverages a fixed real-world data set, active evaluation aims at reducing the labeling cost by adaptively choosing data for human labeling [14, 46]. Active evaluation has also been extended to `AutoEval` [54, 49, 17]. This line of work is orthogonal to the solution proposed in our work, which focuses on adaptively weighting synthetic data rather than adaptively selecting real-world data.

**Game-theoretic statistics.** Game-theoretic statistics builds on the notion of e-value. E-values are measures of evidence that offer several advantages over conventional p-values, including the support of optional continuation [38, 37]. The interpretation of an e-value as the wealth associated with a betting strategy makes it possible to optimize test statistics using tools from online convex optimization, leading to state-of-the-art estimation and testing mechanisms [45, 47, 6]. Moreover, active e-values [49] support the implementation of active evaluation [17, 50].

## B   Algorithm

Algorithm 1 summarizes the overall procedure of `R-AutoEval+`, which reduces to `R-Eval` [47, 6] by setting $\rho_s = 0$ for all $s = 1, ..., S$, and to `R-AutoEval` [20] by setting $\rho_s = 1$ for all $s = 1, ..., S$. In Algorithm 1, we consider the test

$$T_n = \mathbb{1}\left(\max_{i \in \{1,...,n\}} E_i \geq 1/\delta\right), \tag{19}$$

which meets the reliability condition (2) thanks to Ville's inequality [44]. The testing design (19) is standard in testing-by-betting frameworks [47, 6] due to its higher power, as it returns $T_n = 1$ any time the test (5) does.

**Algorithm 1:** `R-AutoEval+`

---

**Input:** human-labeled data $\mathcal{D}_n = \{(X_i, Y_i)\}_{i=1}^n$, unlabeled data $\mathcal{D}_N^{\text{unl}} = \{\tilde{X}_i\}_{i=1}^N$, autoevaluator $f$, target risk level $\alpha$, reliability level $1 - \delta$, reliance factors $\{\rho_s\}_{s=1}^S$ satisfying (9), initial positive weights $\{w_{s,0}\}_{s=1}^S$ satisfying $\sum_{s=1}^S w_{s,0} = 1$, betting algorithm $\mathcal{A}lg$ (see Sec. C)

**Output:** test output $T_n$

---

**initialize** $E_0 \leftarrow 1$, $E_{s,0} \leftarrow 1$; $\lambda_{s,1} = \mathcal{A}lg(\ell_{s,1:0}^f)$ denoting as $\ell_{s,1:m}^f = \{\ell_{s,i}^f\}_{i=1}^m$; $w_{s,1} = w_{s,0}$
for $s = 1, ..., S$

**for** $i = 1, ..., n$ **do**

    **for** $s = 1, ..., S$ **do**

        ◁    *get effective observation $\ell_{s,i}^f$*    ▷

        $\ell_{s,i}^f = \frac{\rho_s}{\lfloor N/n \rfloor} \sum_{i'=\lfloor N/n \rfloor \cdot (i-1)+1}^{\lfloor N/n \rfloor \cdot i} \ell(\tilde{X}_{i'}, f(\tilde{X}_{i'})) + \ell(X_i, Y_i) - \rho_s \cdot \ell(X_i, f(X_i)).$

        ◁    *update $E_{s,i}$*    ▷

        $E_{s,i} \leftarrow E_{s,i-1} \cdot \underbrace{(1 - \lambda_{s,i} \cdot (\ell_{s,i}^f - \alpha))}_{=e_{s,i}}$

        ◁    *update betting variable $\lambda_{s,i+1}$*    ▷

        $\lambda_{s,i+1} = \mathcal{A}lg(\ell_{s,1:i}^f)$

    **end**

    ◁    *update e-value $E_i$*    ▷

    $E_i \leftarrow E_{i-1} \cdot \sum_{s=1}^S w_{s,i} \cdot e_{s,i}$

    ◁    *update weights $w_{s,i+1}$*    ▷

    $w_{s,i+1} = \frac{w_{s,0} \cdot E_{s,i}}{\sum_{s'=1}^S w_{s',0} \cdot E_{s',i}}$ for all $s = 1, ..., S$.

**end**

**return** $\mathbb{1}\big( \max_{i \in \{1,...,n\}} E_i \geq 1/\delta \big)$

## C   Betting Strategy

In this section, we summarize commonly used betting strategies [47, 48], namely Waudby-Smith–Ramdas (WSR) [47] and universal portfolio (UP) [15]. While both schemes show excellent performance empirically, we start with UP, which satisfies assumption A1 in Theorem 2.

In this section, we denote as $\mathcal{A}lg$ a betting algorithm that maps the available past observations into the next betting variable, i.e., at round $i$, given previous observations $q_{1:i-1} = \{q_j\}_{j=1}^{i-1}$ with $q_i \in [m, M]$, a betting algorithm outputs $\lambda_i = \mathcal{A}lg(q_{1:i-1})$.

### C.1   Universal Portfolio (UP)

Recall the definition of $E_n(\lambda) = \prod_{i=1}^n (1 - \lambda(q_i - \alpha))$, which is the e-value (6) with $\lambda_i = \lambda$ for all $i = 1, ..., n$. UP [15] defines the betting variable $\lambda_i$ at round $i$ as

$$\mathcal{A}lg(q_{1:i-1}) = \frac{1}{M - \alpha} \frac{\int_0^1 \lambda E_{i-1}(\lambda/(M-\alpha)) \mathrm{d}F(\lambda)}{\int_0^1 E_{i-1}(\lambda/(M-\alpha)) \mathrm{d}F(\lambda)}, \tag{20}$$

which satisfies the assumption A1 in Theorem 2 (see also [48, 36]). The integrals in (20) are computed by discretizing the continuous domain $[0, 1]$ of $\lambda$ into a uniformly spaced grid of size 10000 in a manner similar to [15, Sec. 8].

Table 2: Computational complexity comparison of the algorithms considered in this work.

| Eval | AutoEval | R-Eval | R-AutoEval | R-AutoEval+ |
|------|----------|--------|------------|-------------|
| $O(n)$ | $O(N)$ | $O(n + nG)$ | $O(nr + nG)$ | $O(Snr + SnG)$ |

## C.2 Waudby-Smith–Ramdas (WSR)

WSR defines the betting variable $\lambda_i$ at round $i$ as [47, 6, 20]

$$\mathcal{A}lg(q_{1:i-1}) = \min\left\{ \frac{c}{M - \alpha}, \sqrt{\frac{2\log(1/\delta)}{n\hat{\sigma}_{i-1}^2}} \right\}, \tag{21}$$

where the empirical standard deviation and empirical mean at round $i - 1$ and at round $j$ are respectively defined as

$$\hat{\sigma}_{i-1}^2 = \frac{\hat{\sigma}_0^2 + \sum_{j=1}^{i-1}(q_j - \hat{\mu}_j)^2}{i}, \text{ and } \hat{\mu}_j = \frac{\hat{\mu}_0 + \sum_{k=1}^{j} q_k}{j + 1} \tag{22}$$

with initial guesses $\hat{\mu}_0 = 1/2$ and $\hat{\sigma}_0^2 = 1/4$ [47, 20]. We set $c = 3/4$ following [47].

## D  Computational Complexity

Here we compare the computational complexities of the algorithms considered in this work. Given both the human-labeled (size $n$) and autoevaluated data (size $N$), mean-based approaches (Eval and AutoEval, which do not come with reliability guarantees), require $O(n)$ (for Eval) and $O(N)$ (for AutoEval) operations for computing the respective average.

The reliable approaches (R-Eval and R-AutoEval) need additional computation steps for updating the e-values as well as the betting strategies (Sec. 2). While updating the e-values at each round requires a single multiplication, the betting strategy may be more complex. For example, especially the universal portfolio (UP) strategy with sublinear regret (Assumption A1) has a complexity of order $O(nG)$ where $G$ is the size of the grid used to approximate the integral operation of UP (Sec. C.1), although this can be decreased by adopting sampling-based approaches [29].

Lastly, R-AutoEval+ requires $S$ times more computation than R-AutoEval due to its consideration of $S$ candidate factors. It is worth emphasizing that, the costs described above are generally negligible with respect to running the autoevaluator (e.g., an LLM judge). The summary of the computation complexities can be found in Table 2.

## E  Additional Experimental Results

While UP betting provably achieves sublinear regret bound (A1 in Theorem 2), WSR betting shows excellent empirical result with significantly lower computational overhead. Furthermore, existing works on R-Eval [47, 6] and R-AutoEval [20] adopted WSR as their betting strategy. Accordingly, here we investigate the impact of the choice of betting strategy between WSR and UP (see Sec. C for the summary of WSR and UP). Note that we have used WSR for Fig. 1 to maintain consistency with prior art, while for all the other figures, we have used UP, which satisfies assumption A1 in Theorem 2. In this section, we provide all the missing figures with the alternative betting strategy choice. After showing all the counterparts of the figures in the main text, we provide additional experiments on LLM quantization for CoQA data set [39] and on constructing two-sided confidence interval for the setting studied in Examples 1 and 2. Lastly, we provide an extended version of Table 1 for LLM reasoning task by accounting a wider range of autoevaluators. We conclude this section by investigating the impact of R-AutoEval+'s hyperparameters such as $S$, $\{w_{s,0}\}_{s=1}^S$, and $\{\rho_s\}_{s=1}^S$, as well as of the ordering of data, on the amount of reasoning tokens that can be saved while ensuring the target 3% accuracy gain.

### E.1  Toy Example

In Fig. 5, we provide the counterpart to Figs. 2 and 3, considering a WSR betting strategy instead of the UP one. It is observed that WSR betting makes the weights of R-AutoEval+ less concentrated around

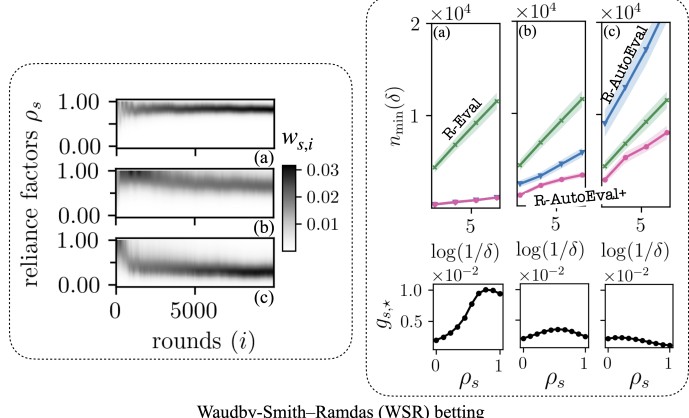

Waudby-Smith–Ramdas (WSR) betting

Figure 5: Same setting with Figs. 2 and 3 but with WSR betting instead of UP betting.

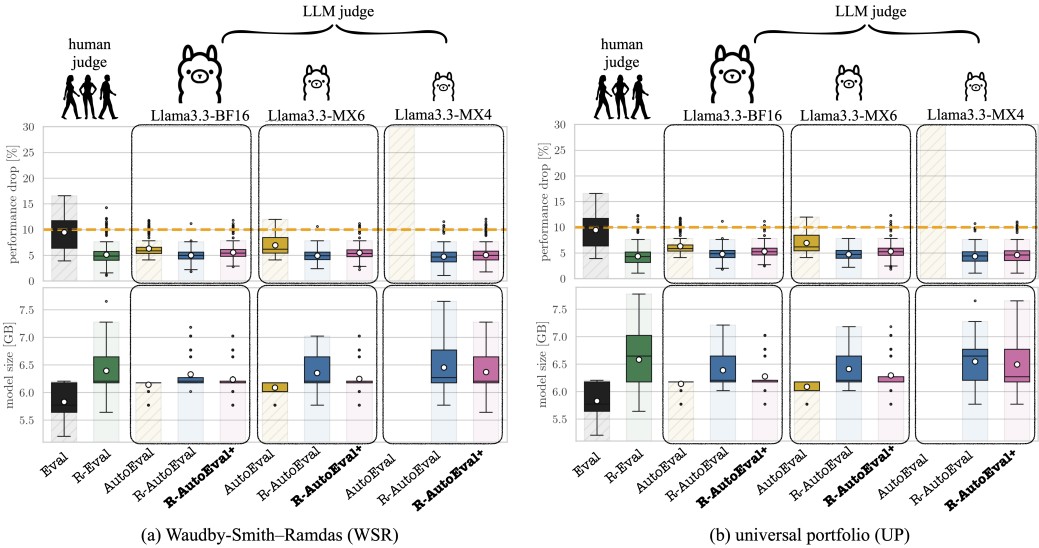

(a) Waudby-Smith–Ramdas (WSR)

(b) universal portfolio (UP)

Figure 6: (a) Box plot for Fig. 1; (b) same setting as (a) but with UP betting strategy (see Sec. C.1).

the optimal reliance factor, although it still outperforms the other two schemes with a substantial margin.

## E.2   LLM Quantization

Fig. 6(a) shows the box plot for Fig. 1. Fig. 1 reports maximum values within the 1.5 IQR range [34]. Fig. 6(b) plots the same box plot but with UP betting instead of WSR betting. Fig. 7(a) is the counterpart of Fig. 4(a) which uses WSR betting in lieu of UP betting. It is observed that even under the WSR betting strategy, `R-AutoEval+` consistently outperforms `R-Eval` and `R-AutoEval`, returning a smaller quantized model.

## E.3   LLM Prompting

Fig. 7(b) is the counterpart of Fig. 4(b) which uses WSR betting in lieu of UP betting. Fig. 8 then shows per-task results of Instruction-Induction task [27] with (a) UP betting and (b) WSR betting. We show 17 tasks among 24 tasks that have more than 2000 examples [27]. Even under the different choice of betting strategy, `R-AutoEval+` outperforms both `R-Eval` and `R-AutoEval`.

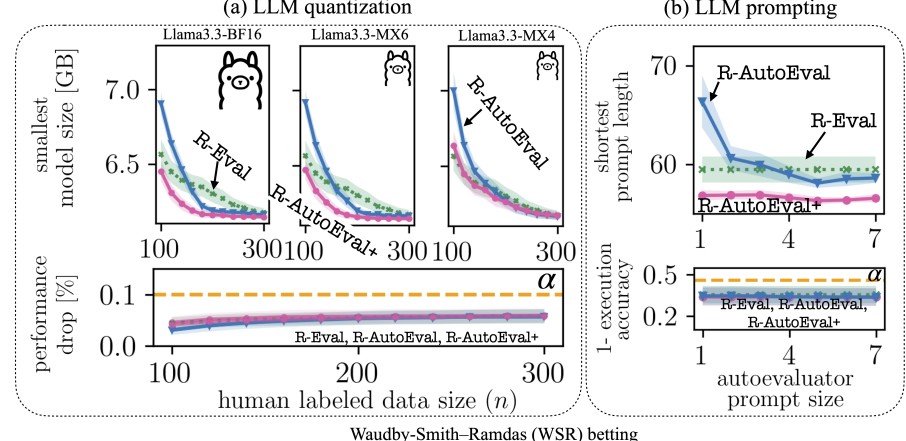

Figure 7: Same setting as Fig. 4 but with WSR betting strategy (see Sec. C.2).

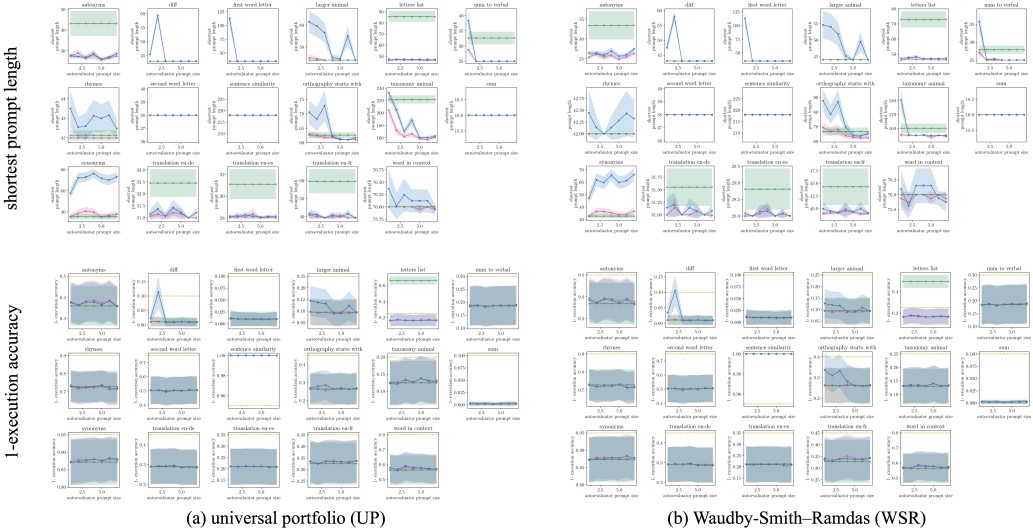

Figure 8: (a) Per-task result for Fig. 4(b); (b) same setting as (a) but with WSR betting strategy.

## E.4 LLM Quantization: CoQA Data Set

We extend the experimental validation of `R-AutoEval+` by considering a different data set for LLM quantization task. Fig. 9 is the counterpart of Fig. 4(a) but with CoQA data set [39] under (a) WSR betting and (b) UP betting. The same trend is observed, confirming the improved efficiency of `R-AutoEval+`.

## E.5 Confidence Interval

Under the same setting as Examples 1 and 2, we consider the construction of two-sided confidence interval for the expected risk $R$ using `R-Eval`, `R-AutoEval`, and `R-AutoEval+`. We follow the *hedged capital process* [47, Sec. 4.4] to first construct $(1 - \delta \cdot \varepsilon)$-upper confidence bound and $(1 - \delta \cdot (1 - \varepsilon))$-lower confidence bound separately to yield their intersection as the two-sided confidence interval. An upper confidence bound can be obtained by finding the largest $\alpha$ for which the respective testing $T_n$ in (5) yields 0; a lower confidence bound can be obtained similarly by considering the observation $q_i^{lb} = m + M - q_i$ in lieu of the observation $q_i$. We set $\varepsilon = 0.5$ and adopt uniform grid of size 10000 to approximately find the largest $\alpha$ with $T_n = 0$. We employ the WSR betting strategy, and set the upper bound of $\lambda_i$ in (21) to $1/(M - m)$ to ensure the monotonicity

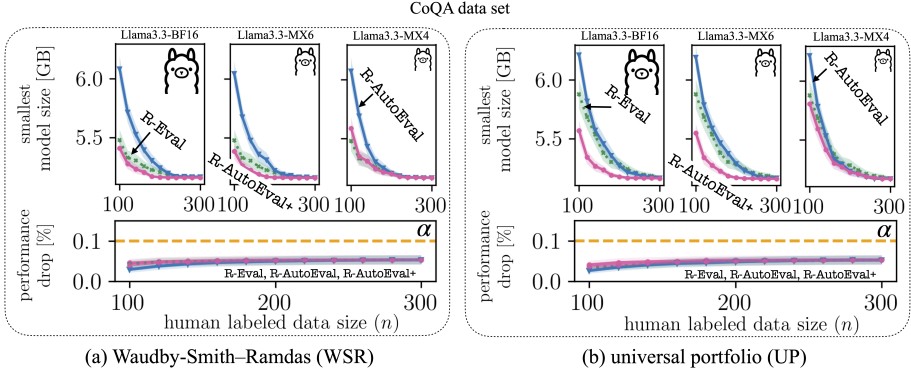

Figure 9: LLM quantization task on CoQA data set [39]. All the other settings are the same with Fig. 4.

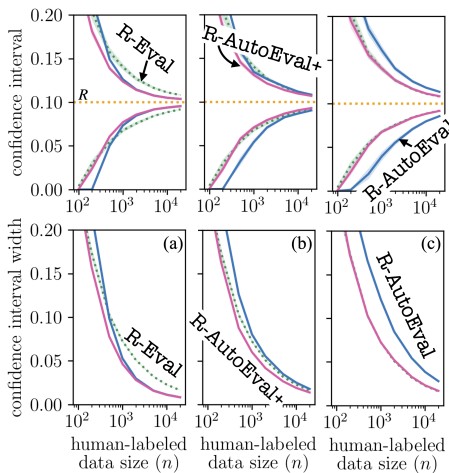

Figure 10: 99.9% two-sided confidence interval for the same setting in Example 2.

of `R-Eval`'s and `R-AutoEval`'s e-values [20, 6]. Fig. 10 shows the corresponding (top) confidence interval and (bottom) its width versus the number of real-world data $n$. Recall from Example 1 and 2 that the quality of synthetic data decreases in the order of (a), (b), and (c). Across all scenarios, `R-AutoEval+` returns confidence intervals that have the smallest size.

## E.6 LLM Reasoning: Wider range of autoevaluators

Table 3 extends Table 1 in the main text by considering additional autoevaluators: BitNet [32], Llama-3 [19], DeepSeek-R1 [25], Qwen3 [51], and GPT4 [1]. The table confirms again the efficiency gain of `R-AutoEval+` as compared to `R-Eval` and `R-AutoEval`; also demonstrating again the benefit of choosing autoevaluator from the different family of the base model to achieve better efficiency for autoeval-based approaches.

## E.7 LLM Reasoning: Sensitivity of testing-by-betting-type algorithms with respect to data ordering

We first recall that the algorithms that builds upon the testing-by-betting framework [47] need to impose some arbitrary ordering of the data to sequentially run the algorithm. Here we investigate the impact of such ordering on the final decision of the algorithms. To this end, we fix the calibration-test data split and only change the ordering of data randomly for 100 times to compute the normalized deviation (standard deviation / mean) of the generated tokens dictated by each algorithm. We consider

Table 3: Selecting the smallest reasoning budget for Qwen3-1.7B that ensures at least 3% accuracy enhancement as compared to its non-reasoning mode, evaluated on GSM8K data set [13]: average number of generated tokens with standard deviation shown within parentheses.

| autoevaluator (accuracy) | R-Eval | R-AutoEval | R-AutoEval+ |
|---|---|---|---|
| BitNet b1.58 (35%) | | 950.27 (152.84) | **942.47 (135.65)** |
| Llama-3.2-1B-Instruct (26%) | | 964.91 (136.55) | **954.63 (145.60)** |
| DeepSeek-R1-Distill-Qwen-1.5B (33%) | | 943.98 (136.38) | **935.58 (130.83)** |
| Llama-3.2-3B-Instruct (66%) | | 900.58 (122.70) | **892.86 (112.10)** |
| DeepSeek-R1-Distill-Qwen-7B (54%) | | 941.66 (126.32) | **926.85 (118.08)** |
| DeepSeek-R1-Distill-Llama-8B (67%) | | 916.34 (129.83) | **900.82 (119.73)** |
| DeepSeek-R1-0528-Qwen3-8B (32%) | 983.34 (137.87) | 986.53 (140.77) | **961.94 (138.98)** |
| Qwen3-32B (82%) | | 1007.45 (150.48) | **941.20 (129.61)** |
| DeepSeek-R1-Distill-Qwen-32B (89%) | | 893.39 (105.13) | **866.22 (80.58)** |
| Llama-3.3-70B-Instruct (89%) | | 854.42 (90.26) | **847.05 (69.21)** |
| DeepSeek-R1-Distill-Llama-70B (88%) | | 863.72 (71.60) | **858.02 (69.59)** |
| GPT-4.1 nano (90%) | | 879.45 (93.95) | **865.28 (72.34)** |
| GPT-4.1 (93%) | | 883.99 (103.00) | **856.13 (70.93)** |

Table 4: Sensitivity of algorithms with respect to data ordering. Normalized deviation (standard deviation / mean) of the generated tokens yielded by each algorithm for a fixed calibration-test data split is shown for each algorithm.

| autoevaluator | R-Eval | R-AutoEval | R-AutoEval+ |
|---|---|---|---|
| GPT4.1 | 8.62% | 3.90% | 6.88% |
| DeepSeek-R1-0528-Qwen3-8B | 8.62% | 6.32% | 6.00% |

GPT-4.1 and DeepSeek-R1-0528-Qwen3-8B as the autoevaluators for the experimental results. As can be seen from Table 4, R-AutoEval and R-AutoEval+ are more robust to the data ordering than R-Eval, which may be possibly attributed to its reduced variability of averaged autoevaluated data (at each round we average $r$ number of autoevaluated examples), while R-AutoEval+ tends to be slightly less robust than R-AutoEval for its adaptive nature that incorporates the data ordering.

### E.8   LLM Reasoning: Impact of the number of candidate factors $S$

We now start investigating the impact of hyperparameters required by R-AutoEval+. We first investigate the impact on the number of candidates, $S$, with candidate factors equally spaced within the interval $[0, 1]$; and with equal initial weights. We report the average number of generated tokens with corresponding standard deviation in a manner similar to Table 1 and 3.

As can be observed in Table 5, as long as the candidate factors includes $\rho = 0$ (R-Eval) and $\rho = 1$ (R-AutoEval), i.e., as long as $S \geq 2$, R-AutoEval+ consistently outperforms both R-Eval and R-AutoEval, with its gain being maximized at $S = 5$, and saturates after $S = 10$.

### E.9   LLM Reasoning: Impact of the values of candidate factors ($\{\rho_s\}_{s=1}^{S}$)

Next, we investigate the impact of the values of each candidate factors by going beyond the equally spaced grid. To this end, we consider Dirichlet distribution with concentration parameter $0.1, 1, 10$ to allocate the values within the $[0, 1]$ interval. Specifically, the cumulative probabilities of the Dirichlet distribution are used as the candidate factors and we keep the first and the last factors to be $0$ and $1$ to

Table 5: Impact of the number of candidate factors ($S$) for R-AutoEval+. The other settings are the same as in Table 1 and 3.

| autoevaluator | $S = 2$ | $S = 5$ | $S = 10$ | $S = 20$ |
|---|---|---|---|---|
| GPT4.1 | 871.67 (16.24) | **854.99 (14.02)** | 856.13 (13.90) | 856.13 (13.90) |
| DeepSeek-R1-0528-Qwen3-8B | 964.41 (27.17) | **961.94 (27.24)** | **961.94 (27.24)** | **961.94 (27.24)** |

Table 6: Impact of the values of candidate factors ($\{\rho_s\}_{s=1}^{S}$) for R-AutoEval+. The other settings are the same as in Table 1 and 3.

| autoevaluator | equal grid | $\text{Dir}(10, ..., 10)$ | $\text{Dir}(1, ..., 1)$ | $\text{Dir}(0.1, ..., 0.1)$ |
|---|---|---|---|---|
| GPT4.1 | **856.13 (13.90)** | **856.13 (13.90)** | 860.91 (15.63) | 873.27 (15.10) |
| DeepSeek-R1-0528-Qwen3-8B | 961.94 (27.24) | 964.22 (26.77) | **960.76 (27.09)** | 966.72 (27.46) |

Table 7: Impact of the initial weights ($\{w_{s,0}\}_{s=1}^{S}$) for R-AutoEval+. The other settings are the same as in Table 1 and 3.

| autoevaluator | equal weight | $\text{Dir}(10, ..., 10)$ | $\text{Dir}(1, ..., 1)$ | $\text{Dir}(0.1, ..., 0.1)$ |
|---|---|---|---|---|
| GPT4.1 | **856.13 (13.90)** | **856.13 (13.90)** | **856.13 (13.90)** | 856.68 (12.31) |
| DeepSeek-R1-0528-Qwen3-8B | 961.94 (27.24) | **959.21 (27.04)** | 962.41 (26.52) | 969.50 (25.59) |

ensure the efficiency guarantee as per our Theorem 3. An increased concentration parameter makes samples from the Dirichlet distribution closer to the uniform distribution, and hence the grid becomes closer to the equally spaced grid. Table 6 shows the preference for setting the grid with equal spacing.

### E.10 LLM Reasoning: Impact of the weights associated with the candidate factors ($\{w_{s,0}\}_{s=1}^{S}$)

The last remaining degree of freedom is given by the choice of the weights associated with each candidate factor. To this end, we go beyond the uniform distribution and consider again the Dirichlet distribution to allocate the weights for the candidate factors. Table 7 confirms again the preference for equal weighting.

## F Proofs

Here we provide proofs of Lemma 1 and Theorem 3.

### F.1 Proof of Lemma 1

This proof follows as in [16, Sec. III] (see also [18, Sec. A] for a more general version).

*Proof.* We first rewrite the e-value $E_{s,i}$ in (12) in an exponential form as

$$E_{s,i} = \exp\left( \sum_{j=1}^{i} \underbrace{\log\left( (1 - \lambda_{s,j}(\ell_{s,j}^{f} - \alpha)) \right)}_{=g_{s,j}} \right), \tag{23}$$

where $g_{s,j}$ is the increment of log-e-value $\log E_{s,j}$ at round $j$. We will further simplify the notation by writing $g_{s,1:i}$ as the summation of logarithmic increments up to round $i$, i.e., $g_{s,1:i} = \sum_{j=1}^{i} g_{s,j}$. The update rule (13) for the weight $w_{s,i}$ can then be rewritten as

$$w_{s,i} = \frac{w_{s,0} \cdot \exp\left( g_{s,1:i-1} \right)}{\sum_{s'=1}^{S} w_{s',0} \cdot \exp\left( g_{s',1:i-1} \right)}. \tag{24}$$

Now, by unfolding the product term in (11) as

$$E_n = \frac{\sum_{s=1}^{S} w_{s,0} \cdot \exp\left( g_{s,1:1} \right)}{\sum_{s=1}^{S} w_{s,0} \cdot 1} \times \cdots \times \frac{\sum_{s=1}^{S} w_{s,0} \cdot \exp\left( g_{s,1:n} \right)}{\sum_{s=1}^{S} w_{s,0} \cdot \exp\left( g_{s,1:n-1} \right)}, \tag{25}$$

we have

$$E_n = \sum_{s=1}^{S} w_{s,0} \cdot \exp\left( g_{s,1:n} \right) = \sum_{s=1}^{S} w_{s,0} E_{s,n}. \tag{26}$$

For any positive weights $\{w_{s,0}\}_{s=1}^S$, it follows that

$$E_n \geq w_{s,0}E_{s,n} \quad \text{for all } s = 1, ..., S, \tag{27}$$

from which we conclude

$$\max_{s=1,...,S} \log E_{s,n} - \log E_n \leq \max_{s\in\{1,...,S\}} \log\left(\frac{1}{w_{s,0}}\right). \tag{28}$$

$\square$

## F.2 Proof of Theorem 3

We start this subsection with proof sketch of Theorem 3 that builds upon [48, B.4]. This is followed by a series of useful properties of the R-AutoEval+'s e-value (11), which are instrumental for proving Theorem 3. The full proof then follows the proof technique in [48, B.4].

### F.2.1 Proof sketch

Waudby-Smith et al. [48] showed that the sample complexity of testing (5) given some e-value $E_n$ of the form (6) can be studied by investigating the simpler e-value with optimal constant betting strategy $\lambda_\star$. Recall that in Sec. 3.2.1, we have already defined such simpler e-value, and we repeat its definition here again for convenience

$$E_n(\lambda_\star) = \prod_{i=1}^n \left(1 - \lambda_\star(q_i - \alpha)\right). \tag{29}$$

The rationale for this simplification is as follows: (*i*) The simpler e-value cannot be too large as compared to the actual e-value obtained using betting strategy that satisfies assumption A1 in Theorem 2 (sublinear regret); (*ii*) the actual e-value also cannot be too large as compared to the simpler e-value (so called numeraire property, see below Lemma 2), hence one can instead study the sample complexity of such simpler e-value in order to characterize the actual sample complexity of R-Eval.

That is to say, as long as we can also find a simpler e-value with a bounded deviation from R-AutoEval+'s e-value $E_n$ in (11), we can then study the corresponding sample complexity instead. As can be anticipated from Theorem 3, such simpler e-value is described as

$$E_{\bar{s},n}(\lambda_{\bar{s},\star}) = \prod_{i=1}^n \left(1 - \lambda_{\bar{s},\star}(\ell_{\bar{s},i}^f - \alpha)\right), \tag{30}$$

where $\bar{s}$ is the index that maximizes the expected increment of the $s$-th log-e-value, i.e.,

$$\bar{s} = \arg\max_{s=1,...,S} g_{s,\star}. \tag{31}$$

While intuitively R-AutoEval+'s e-value has a bounded deviation from $E_{\bar{s},n}(\lambda_{\bar{s},\star})$, one subtlety here is that, the best index $\bar{s}$ might not be equal with the one that is chosen at hindsight, i.e., the following event can happen

$$\arg\max_{s=1,...,S} g_{s,\star} \neq \arg\max_{s=1,...,S} E_{s,n} \tag{32}$$

with non-negligible probability. To overcome this issue, we focus on the upper bound of the sample complexity by considering a *deterministically* no larger e-value $E_n^+$ as compared to R-AutoEval+'s e-value, defined as follows

$$E_n^+ = \sum_{s=1}^S w_{s,0} \min\{E_{s,n}, E_{\bar{s},n}\} \leq \sum_{s=1}^S w_{s,0}E_{s,n} \overset{(26)}{=} E_n. \tag{33}$$

Accordingly, we denote the sample complexity of $E_n^+$ as

$$n_{\min}^+(\delta) = \mathbb{E}[\min\{n : E_n^+ \geq 1/\delta\}|R \leq \alpha] \tag{34}$$

which is no smaller than $n_{\min}^{\texttt{R-AutoEval+}}$ due to (33). We then show the upper bound of $n_{\min}^+(\delta)$, which will prove Theorem 3.

### F.2.2 Useful properties

We first summarize the useful properties of $E_n^+$, which bounds its amount of deviation with respect to the simpler e-value $E_{\bar{s},n}(\lambda_{\bar{s},\star})$. We first discuss the *regret*, which dictates the lower bound deterministically, then discuss the *suboptimality ratio* which captures its upper bound probabilistically.

**Regrets:** The regret of $E_n^+$ as compared to the one that chooses the constant betting at hindsight with index chosen as $\bar{s}$ can be formally defined as

$$
r_k^+ = \max_{\lambda \in [0, 1/(1+\rho_{\bar{s}} - \alpha))} \log E_{\bar{s},k}(\lambda) - \log E_k^+
$$

$$
= \max_{\lambda \in [0, 1/(1+\rho_{\bar{s}} - \alpha))} \log E_{\bar{s},k}(\lambda) - \log E_{\bar{s},k} + \log E_{\bar{s},k} - \log E_k^+
$$

$$
\overset{(a)}{=} \underbrace{\max_{\lambda \in [0, 1/(1+\rho_{\bar{s}} - \alpha))} \log E_{\bar{s},k}(\lambda) - \log E_{\bar{s},k}}_{= \mathrm{regret}_{\bar{s},k}^{\mathrm{bet}}} + \log E_{\bar{s},k} - \log \underbrace{\sum_{s=1}^{S} w_{s,0} \min\{E_{s,k}, E_{\bar{s},k}\}}_{\geq w_{\bar{s},0} E_{\bar{s},k}}
$$

$$
\leq \mathrm{regret}_{\bar{s},k}^{\mathrm{bet}} + \log E_{\bar{s},k} - \log(w_{\bar{s},0} E_{\bar{s},k}) = \mathrm{regret}_{\bar{s},k}^{\mathrm{bet}} + \log(1/w_{\bar{s},0}). \tag{35}
$$

where (a) is due to the definition of $E_k^+$ in (33), and we have denoted as $\mathrm{regret}_{s,n}^{\mathrm{bet}}$ the regret of betting associated with the $s$-th e-value $E_{s,i}$ in (12) at round $n$, which is sublinear under the choice of betting strategy as per assumption A1 in Theorem 2, e.g., universal portfolio strategy (see Appendix C). Accordingly, under assumption A1 in Theorem 2, we can conclude that $r_k^+$ is sublinear, i.e.,

$$
r_k^+ = o(n). \tag{36}
$$

**Suboptimality ratio:** The probabilistic upper bound of e-values can be captured by the expected ratio with respect to the simpler e-values. We first summarize the known results of R-Eval [48].

**Lemma 2** (R-Eval's suboptimality ratio [48, Lemma 5.3]). *Consider an e-value of the form (6) with some arbitrary betting strategy $\lambda_i$. Under the alternative hypothesis $\mathcal{H}_1 : R \leq \alpha$, the following property holds at any round $n$*

$$
\mathbb{E}\left[\frac{E_n}{E_n(\lambda_\star)}\right] \leq 1. \tag{37}
$$

We then extend the above property to $E_n^+$ in (33).

**Lemma 3** ($E_n^+$'s suboptimality ratio). *Consider the e-value $E_n^+$ in (33) with some arbitrary betting strategy $\{\lambda_{s,i}\}_{s=1}^{S}$. Under the alternative hypothesis $\mathcal{H}_1 : R \leq \alpha$, the following property holds at any round $n$*

$$
\mathbb{E}\left[\frac{E_n^+}{E_{\bar{s},n}(\lambda_{\bar{s},\star})}\right] \leq 1. \tag{38}
$$

*Proof.* We first recall $E_n^+ = \sum_{s=1}^{S} w_{s,0} \min\{E_{s,n}, E_{\bar{s},n}\}$ from (33). We then have

$$
\mathbb{E}\left[\frac{E_n^+}{E_{\bar{s},n}(\lambda_{\bar{s},\star})}\right] \leq \mathbb{E}\left[\frac{\sum_{s=1}^{S} w_{s,0} E_{\bar{s},n}}{E_{\bar{s},n}(\lambda_{\bar{s},\star})}\right] = \mathbb{E}\left[\frac{E_{\bar{s},n}}{E_{\bar{s},n}(\lambda_{\bar{s},\star})}\right] \leq 1. \tag{39}
$$

where the last inequality is due to Lemma 2 by noting that $\bar{s}$ is a fixed variable. $\qquad\square$

### F.2.3 Full proof

We now show the upper bound of R-AutoEval+. Most of the steps are identical with the proof technique of [48, B.4] that uses regret and suboptimality ratio to get an upper bound of the sample complexity. We just need to take into account for the regret and suboptimality ratio that are tailored for R-AutoEval+ as defined in the previous subsection.

**Upper bounding** $n_{\min}^{\texttt{R-AutoEval+}}(\delta)$**.** As discussed earlier, we focus on the upper bound of $n_{\min}^+(\delta)$ (see (34)).

Waudby-Smith et al. [48, B.4] started the proof by first showing that the expected stopping time $\mathbb{E}[\tau]$ given the stopping time $\tau$ of a random process $(W_i)_{i=1}^{\infty}$ defined as $\tau = \inf\{n \in \mathbb{N} : W_n \geq 1/\delta\}$, can be simplified as

$$\mathbb{E}[\tau] \leq m + 1 + \sum_{k=m}^{\infty} \Pr[|k^{-1} \log W_k - g| \geq c], \tag{40}$$

for some constant $g$, where the variables $c$ and $m$ are respectively defined as

$$c = \frac{\beta}{1+\beta} g \quad \text{and} \quad m = \left\lceil \frac{\log(1/\delta)}{g - c} \right\rceil, \tag{41}$$

given some $\beta \in (0,1)$.

We now start our proof by taking $E_i^+$ and $g_{\bar{s},\star}$ in lieu of $W_i$ and $g$ from (40), i.e.,

$$n_{\min}^+(\delta) \leq m + 1 + \underbrace{\sum_{k=m}^{\infty} \Pr\left[|k^{-1} \log E_k^+ - g_{\bar{s},\star}| \geq c,\right.}_{(\square)} \tag{42}$$

with the variables $c$ and $m$ being redefined as

$$c = \frac{\beta}{1+\beta} g_{\bar{s},\star} \quad \text{and} \quad m = \left\lceil \frac{\log(1/\delta)}{g_{\bar{s},\star} - c} \right\rceil. \tag{43}$$

Following the next step of Waudby-Smith et al. [48, B.4], we upper bound $(\square)$ as

$$\square \leq \underbrace{\sum_{k=m}^{\infty} \Pr\left[|k^{-1} \log E_k^+ - k^{-1} \log E_{\bar{s},k}(\lambda_{\bar{s},\star})| \geq c/2\right]}_{(\dagger)}$$

$$+ \underbrace{\sum_{k=m}^{\infty} \Pr\left[|k^{-1} \log E_{\bar{s},k}(\lambda_{\bar{s},\star}) - g_{\bar{s},\star}| \geq c/2\right]}_{(\dagger\dagger)}. \tag{44}$$

Note that each term in (44) measures the tail probability associated with the deviation between $(\dagger)$ : $E_k^+$'s log-e-value and the one with optimal constant betting/weighting $(\dagger\dagger)$ : average log-e-value and the expected log-e-value.

We first bound $(\dagger)$. The $k$-th summand of $(\dagger)$ can be bounded as follows denoting as $\Delta_k = k^{-1} \log(E_k^+/E_{\bar{s},k}(\lambda_{\bar{s},\star}))$:

$$\Pr[|\Delta_k| \geq c/2] \leq \Pr[\Delta_k \geq c/2] + \Pr[-\Delta_k \geq c/2]$$
$$= \Pr\left[\log(E_k^+/E_{\bar{s},k}(\lambda_{\bar{s},\star})) \geq kc/2\right]$$
$$+ \Pr[k^{-1} \log E_{\bar{s},k}(\lambda_{\bar{s},\star}) - k^{-1} \log E_k^+ \geq c/2]$$
$$\leq \exp(-kc/2) + \mathbb{1}\left(k^{-1} r_k^+ \geq c/2\right), \tag{45}$$

where the first part of the inequality is due to Lemma 3; the second part of the inequality is due to the sublinear regret (36).

The rest is identical with [48, B.4], although we proceed the proof to make the paper self-contained. The term $(\dagger)$ can be further bounded by noting $\sum_{k=m-1}^{\infty} \exp(-kc/2) \leq 2(1+\beta)\delta^{\beta/2}/\beta g_{\bar{s},\star}$ as

$$(\dagger) \leq 1 + \frac{2(1+\beta)\delta^{\beta/2}}{\beta g_{\bar{s},\star}} + \sum_{k=m}^{\infty} \mathbb{1}\left(k^{-1} r_k^+ \geq c/2\right) \tag{46}$$

where the third term is finite if and only if $r_k^+$ is sublinear.

The remaining term, $(\dagger\dagger)$, can be bounded using concentration inequality, in particular Chebyshev-Nemirovski inequality [48, Lemma B.1]

$$(\dagger\dagger) \leq 1 + \frac{2^{\sigma} \nu_{\sigma,\star}}{c^{\sigma/2} m^{\sigma/2-1}(\sigma/2-1)}, \tag{47}$$

where we denote the $s^{\text{th}}$ central moment as $\nu_{\sigma,\star} = \mathbb{E}[|\log(1 - \lambda_{\bar{s},\star}(\ell^f_{\bar{s},i} - \alpha)) - g_{\bar{s},\star}|^\sigma | R \le \alpha]$, which is finite as per the assumption A2 made in Theorem 3.

Putting everything together, we get the upper bound for any $\beta \in (0,1)$ as

$$n^+_{\min}(\delta) \le 4 + \frac{(1+\beta)\log(1/\delta)}{g_{\bar{s},\star}} + \frac{2(1+\beta)\delta^{\beta/2}}{\beta g_{\bar{s},\star}}$$
$$+ \frac{2^\sigma \nu_{\sigma,\star}}{\beta(\sigma/2 - 1)}\left(\frac{1+\beta}{g_{\bar{s},\star}} + \frac{1}{\log(1/\delta)}\right) + \sum_{k=m}^\infty \mathbb{1}(k^{-1}r^+_k \ge c/2). \tag{48}$$

In the limit $\delta \to 0^+$, and when considering the ratio $n^+_{\min}(\delta)/\log(1/\delta)$, one can make the factor $(1+\beta)$ arbitrarily close to 1 in a manner similar to [48], which concludes the proof of Theorem 3. $\quad\square$

