# OpenReview forum: "Adaptive Prediction-Powered AutoEval with Reliability and Efficiency Guarantees"
_NeurIPS.cc/2025/Conference — NeurIPS 2025 spotlight_

### Official Review · Reviewer_dCNW · 2025-06-30

**Clarity:** 3
**Significance:** 2
**Originality:** 3
**Rating:** 4
**Confidence:** 3

**Summary:**

The paper introduces a new method called R-AutoEval+ that uses no more samples than the previous autoeval methods and has reliability guarantees. It evaluates a method or model using both human labeled data and auto evaluators (e.g. LLMs-as-judges). The coefficient $\rho$ determining reliance on the auto evaluator is iteratively computed, with one iteration involving one human labeled and many autoeval labeled samples. The theorems have learning theory flavour, mostly focusing on sample efficiency bounds under which the method is reliable i.e. doesn’t overestimate the risk with probability at least $1-\delta$. The main methods used are testing by betting and e-values and PPI++.

**Questions:**

1. In practice how do you deal with the probably correct result i.e. giving the correct answer with probability greater than $1-\delta$? What happens if the prediction is wrong?
2. Have authors compared the introduced method with some other evaluation methods (apart from R-Eval, R-AutoEval and others discussed in the paper)?
3. Line 168, should it be R-Eval instead of AutoEval?

**Ethical Concerns:**

["NO or VERY MINOR ethics concerns only"]

**Final Justification:**

Authors addressed most of my follow-up questions, so I stuck with my original score of 4.

**Limitations:**

Doesn’t discuss what happens in practice when the method returns an incorrect prediction (with probability at most $\delta$).

**Quality:**

4

**Strengths And Weaknesses:**

Strengths:
The paper very nicely and gradually introduces the prior methods, which makes it so much easier to follow. The notation is clear and examples were given. The theorems are clearly stated. The method is tested on two different real-world evaluation problems, evaluating the best (smallest model) “good enough” quantization, and the shortest reliable prompt search.

Weaknesses:
Not sure if significance is well justified. For example, in Figure 9a), it seems like R-AutoEval and R-Eval are very similar. It is hard to know whether this will be a widely used evaluation method in the future.

Seems like the authors renamed the method in [17] to R-AutoEval without mentioning the original name.

---

> ### Author Rebuttal · Authors · 2025-07-31
>
> Thank you for the constructive comments. Below, we provide a point-by-point response to each of the reviewer's comments.
>
> 1. _Not sure if significance is well justified. For example, in Figure 9a), it seems like R-AutoEval and R-Eval are very similar. It is hard to know whether this will be a widely used evaluation method in the future._
>
> > The fact that in some cases using auto evaluator (e.g., LLMs-as-judges) may bring no gain as compared to a vanilla approach is a fundamental limitation of the autoevaluation framework. When the autoevaluator is of insufficient quality and/or the problem at hand is easy enough to be sufficiently analyzed by the human-labeled data, existing approaches such as [17] suffer from performance degradation by utilizing autoevaluators. This is exactly the behavior observed in Figure 9a, where R-AutoEval [17] (blue line) shows worse performance than R-Eval (green).
>
> > However, the benefits are evident when the autoevaluator is sufficiently accurate. For instance, the results for TriviaQA data set (Fig. 4(a), Fig. 7(a)) show that unless the autoevaluator is heavily quantized (i.e., BF16 and MX6), autoevaluator can help finding smaller quantized models. Furthermore,  the results on Instruction-Induction task results in Fig. 4(b), Fig. 7(b) show that unless autoevaluator is operated on low quality prompts (few number of in-context samples), autoevaluator can help finding shorter prompts for the base models.
>
>
> > To further validate the significance of the proposed method, we have carried out additional experiments on more challenging LLM tasks that solves math problems. Specifically, in light of the growing attention on test-time scaling of LLMs [Snell et al., 2024, Muennighoff et al., 2025], we applied our framework to the problem of identifying the minimum amount of additional computation that ensures desired performance gain. We adopted Qwen3-1.7 B as the base model as it provides both non-reasoning and reasoning modes [Yang et al., 2025]; and we considered the task of solving math problems (GSM8K) [Cobbe et al., 2021]. We allow the computation budget for the reasoning mode to vary between 128 to 1280 tokens, and our goal is to find the minimum reasoning budget that achieves at least extra 3% accuracy gain as compared to the non-reasoning mode. The table below shows the amount of test-time computation in the unit of tokens (i.e., the number of generated tokens to answer the math question) that can be saved by R-AutoEval+ as compared to R-Eval (4th column) and R-AutoEval (5th column) for autoevaluators with different sizes, accuracies on GSM8K. We set \$n=1000, N=4000, \delta=0.1$, and report the averaged output from 100 independent runs. We omit the accuracy gain results as all the schemes always achieve at least the target accuracy gain 3%. The table confirms again the efficiency and validity of the proposed framework.
>
> | autoevaluator spec (model name) | autoevaluator spec (model size) | autoevalator spec (acc) | test-time computation gain of R-AutoEval+ over R-Eval (num. tokens)  | test-time computation gain of R-AutoEval+ over R-AutoEval (num. tokens) |
> | :---: | :---: | :---: | :---: | :---: |
> | BitNet b1.58 | 0.85 B | 0.35 | 40.87 | 7.80 |
> | Llama-3.2-3B-Instruct  | 3.21 B | 0.66 | 90.48 | 7.72 |
> | Qwen3-32B | 32.8 B | 0.82 | 42.14 | 66.25 |
> | DeepSeek-R1-Distill-Qwen-32B | 32.8 B | 0.89 | 117.12 | 27.17 |
> | GPT-4.1 | not available | 0.9258 | 127.21 | 27.86 |
>
> **Table 1: LLM test-time scaling experiment for math problem (GSM8K)**
>
>
> 2. _Seems like the authors renamed the method in [17] to R-AutoEval without mentioning the original name._
>
> >  Thank you for pointing this out. The original method in [17] was dubbed “SS-RCPS”, and we have introduced a new name for the consistency. We will revise the manuscript by mentioning the original name when we first introduce [17].
>
> 3. _In practice how do you deal with the probably correct result i.e. giving the correct answer with probability greater than \$1-\delta$? What happens if the prediction is wrong?_
>
> > Wrong predictions are inevitable due to the randomness associated with the realization of the calibration data (see standard PAC-type guarantees). That said,   even in such worst-case settings, R-AutoEval+ can still yield close-to-safe decisions with a minimal violation rate. This can be seen from Fig. 1, where we have reported the worst case within 1.5 interquartile range (IQR) range. Furthermore we have provided the full box plots in Fig. 6 (Appendix D). There, we can observe the attained worst-case performance. The tables below (from high-quality autoevaluator to low-quality autoevaluator) confirm that R-AutoEval and R-AutoEval achieves worst risk lower than 0.125 which corresponds to \$1.25\alpha$ unlike R-Eval that suffers from worst risk near 0.14 (\$1.4\alpha$). Furthermore, the actual violation ratios of R-AutoEval and R-AutoEval+ are lower than R-Eval up to 1% (while all of them being much lower than the target \$\delta=10$%), demonstrating that autoevaluation can provide further benefit in terms of managing such extreme cases.
>
> |  | Eval | R-Eval | AutoEval | R-AutoEval | R-AutoEval+ |
> | --- | --- | --- | --- | --- | --- |
> | violation rate | 51.6% | 2.6% |7.4% | 0.2% |1.4% |
> | worst risk | 0.170 | 0.143 | 0.118 | 0.112 | 0.118 |
>
> **Table 2: Worst-case analysis for LLM quantization, autoeval=Llama3.3-BF16 (Fig. 6(a))**
>
> |  | Eval | R-Eval | AutoEval | R-AutoEval | R-AutoEval+ |
> | --- | --- | --- | --- | --- | --- |
> | violation rate | 51.6% | 2.6% |14.6% | 0.4% |1% |
> | worst risk | 0.170 | 0.143 | 0.120 | 0.106 | 0.112 |
>
> **Table 3: Worst-case analysis for LLM quantization, autoeval=Llama3.3-MX6 (Fig. 6(a))**
>
> |  | Eval | R-Eval | AutoEval | R-AutoEval | R-AutoEval+ |
> | --- | --- | --- | --- | --- | --- |
> | violation rate | 51.6% | 2.6% |100% | 0.8% |1.6% |
> | worst risk | 0.170 | 0.143 | 0.722 | 0.115 | 0.125 |
>
> **Table 4: Worst-case analysis for LLM quantization, autoeval=Llama3.3-MX4 (Fig. 6(a))**
>
> 4. _Have authors compared the introduced method with some other evaluation methods (apart from R-Eval, R-AutoEval and others discussed in the paper)?_
>
> > In the manuscript, we have focused on comparing our methods with Eval and AutoEval (mean-based approach, no reliability guarantees) and R-Eval and R-AutoEval (non-asymptotic reliability guarantees), since they are the state-of-the-art evaluation methods offering statistical validity. For reference, we have now also considered a method that only provides asymptotic reliability guarantees [Boyeau et al., 2024, Fisch et al., 2024] based on the central limit theorem. For the GSM8K data set, we have observed that its violation ratio is 0.12, which is higher than the target level 0.1.  If the work gets accepted, we will further compare our scheme with other  asymptotic benchmarks.
>
> 5. _Line 168, should it be R-Eval instead of AutoEval?_
>
> > You are correct! We will fix the typo, thank you.
>
> #### **References**
> #### [Boyeau et al., 2024] Pierre Boyeau, Anastasios N Angelopoulos, Nir Yosef, Jitendra Malik, and Michael I Jordan. Autoeval done right: Using synthetic data for model evaluation. arXiv preprint arXiv:2403.07008, 2024.
> #### [Fisch et al., 2024] Adam Fisch, Joshua Maynez, R Hofer, Bhuwan Dhingra, Amir Globerson, and William W Cohen. Stratified prediction-powered inference for effective hybrid evaluation of language models. Advances in Neural Information Processing Systems, 37:111489–111514, 2024.

---

> > ### Comment · Reviewer_dCNW · 2025-08-07
> >
> > Thank you authors for the additional clarification! I will keep my acceptance score of 4.

---

### Official Review · Reviewer_UVVV · 2025-07-01

**Clarity:** 3
**Significance:** 3
**Originality:** 3
**Rating:** 4
**Confidence:** 1

**Summary:**

The authors provide a framework for automatic evaluation for AI models, where selecting from multiple candidate responses is necessary. They introduce a framework that provides guarantees on the evaluation performance, with enhanced sample efficiency over conventional methods. Empirically the show the reliability and efficiency of their framework.

**Questions:**

1. Have you done a study of your approach over model size as the reliability of LLM as judge may change with respect to model parameters?
2. Have you looked at scaling properties over risk with respect to the amount of human and synthetic data that is used?
3. Would it be possible to evaluate this framwork in a larger scale setting as the ones decribed above?

**Ethical Concerns:**

["NO or VERY MINOR ethics concerns only"]

**Final Justification:**

I do provide a score of 4 for this evaluation but due to this paper being out of domain from my expertise, I will keep my score conservative towards acceptance. The authors satisfied my empirical concerns in the rebuttal.

**Limitations:**

yes

**Quality:**

3

**Strengths And Weaknesses:**

Strengths
- Theoretically motivated framework with better sample complexity than conventional approaches
- The guarentee provided by this approach is not asymptotic unlike prior work, yielding narrower confidence intervals
- Adaptive weighting of synthetic and real data based on the confidence of the autoevaluator is a novel contribution and leads to better empirical performance

Weaknesses
- Empirically validate their approach on toyish LLM tasks: Quantization and Prompt Shortening tasks. Could be useful to perform a broader evaluation on evaluation of an LLM on a broad range of topics (e.g math, coding, general QA)
- Evaluation limited to a single family of models and judge/autoevaluator (Llama family of models). Could benefit from a broader evaluation over different model families and sizes to see the effect of performance of the strength of the autoevaluator.

---

> ### Author Rebuttal · Authors · 2025-07-31
>
> Thank you for the constructive comments. Below, we provide a point-by-point response to each of the reviewer's comments.
>
> 1) _Empirically validate their approach on toyish LLM tasks: Quantization and Prompt Shortening tasks. Could be useful to perform a broader evaluation on evaluation of an LLM on a broad range of topics (e.g math, coding, general QA)_
>
> > During the rebuttal, we have carried out additional experiments on more challenging LLM tasks. In light of the growing attention on test-time scaling of LLMs [Snell et al., 2024, Muennighoff et al., 2025], we applied our framework to the problem of identifying the minimum amount of additional computation that ensures desired performance gain. We adopted Qwen3-1.7 B as the base model as it provides both non-reasoning and reasoning modes [Yang et al., 2025]; and we considered the task of solving math problems (GSM8K) [Cobbe et al., 2021]. We allow the computation budget for the reasoning mode to vary between 128 to 1280 tokens, and our goal is to find the minimum reasoning budget that achieves at least extra 3% accuracy gain as compared to the non-reasoning mode. The table below shows the amount of test-time computation in the unit of tokens (i.e., the number of generated tokens to answer the math question) that can be saved by R-AutoEval+ as compared to R-Eval (4th column) and R-AutoEval (5th column) for autoevaluators with different sizes, accuracies on GSM8K. We set \$n=1000, N=4000, \delta=0.1$, and report the averaged output from 100 independent runs. We omit the accuracy gain results as all the schemes always achieve at least the target accuracy gain 3%. The table confirms again the efficiency and validity of the proposed framework.
>
> | autoevaluator spec (model name) | autoevaluator spec (model size) | autoevalator spec (acc) | test-time computation gain of R-AutoEval+ over R-Eval (num. tokens)  | test-time computation gain of R-AutoEval+ over R-AutoEval (num. tokens) |
> | :---: | :---: | :---: | :---: | :---: |
> | BitNet b1.58 | 0.85 B | 0.35 | 40.87 | 7.80 |
> | GPT-4.1 | not available | 0.9258 | 127.21 | 27.86 |
>
> **Table 1: LLM test-time scaling experiment for math problem (GSM8K)**
>
> 2) _Evaluation limited to a single family of models and judge/autoevaluator (Llama family of models). Could benefit from a broader evaluation over different model families and sizes to see the effect of performance of the strength of the autoevaluator._
>
> > To address the reviewer’s point, we have considered autoevaluators with different sizes, accuracies on GSM8K data set, whose results and the corresponding test-time computation that can be saved by R-AutoEval+ as compared to R-Eval and R-AutoEval are shown in the table below.
>
> | autoevaluator spec (model name) | autoevaluator spec (model size) | autoevalator spec (acc) | test-time computation gain of R-AutoEval+ over R-Eval (num. tokens)  | test-time computation gain of R-AutoEval+ over R-AutoEval (num. tokens) |
> | :---: | :---: | :---: | :---: | :---: |
> | BitNet b1.58 | 0.85 B | 0.35 | 40.87 | 7.80 |
> | Llama-3.2-1B-Instruct  | 1.24 B | 0.26 | 28.71 | 10.28 |
> | DeepSeek-R1-Distill-Qwen-1.5B | 1.78 B | 0.33 | 47.76 | 8.4 |
> | Llama-3.2-3B-Instruct  | 3.21 B | 0.66 | 90.48 | 7.72 |
> | DeepSeek-R1-Distill-Qwen-7B | 7.62 B | 0.54 | 56.49 | 14.81 |
> | DeepSeek-R1-Distill-Llama-8B | 8.03 B | 0.67 | 82.52 | 15.52 |
> | DeepSeek-R1-0528-Qwen3-8B | 8.19 B | 0.32 | 21.40 | 24.59 |
> | Qwen3-32B | 32.8 B | 0.82 | 42.14 | 66.25 |
> | DeepSeek-R1-Distill-Qwen-32B | 32.8 B | 0.89 | 117.12 | 27.17 |
> | Yi-1.5-34B-Chat | 34.4 B | 0.57 | 74.05 | 5.95 |
> | Llama-3.3-70B-Instruct | 70.6 B | 0.89 | 136.29 | 7.37 |
> | DeepSeek-R1-Distill-Llama-70B | 70.6 B | 0.88 | 125.32 | 5.70 |
> | GPT-4.1 nano | unknown | 0.90 | 118.06 | 14.17 |
> | GPT-4.1 | unknown | 0.93 | 127.21 | 27.86 |
>
> **Table 2: Broad choice of the autoevaluators**
>
> > It is generally observed that choosing the autoevaluator from the same family of the model substantially reduces the gain of autoeval-based approaches. For instance, Llama-3.2-3B-Instruct autoevaluator achieves much lower accuracy – 0.16 – than Qwen3-32B autoevaluator but it can save much more reasoning budget --  48.34 tokens. This demonstrates the importance of the model choice as the reviewer correctly anticipated. We believe that such behavior is related to the positive feedback of LLM judges within the same family [Zheng et al., 2023] also known as preference leakage [Li et al., 2025], which makes the bias correction (8) a more challenging task. We will summarize this behavior in the updated draft if this work gets accepted.
>
> 3) _Have you done a study of your approach over model size as the reliability of LLM as judge may change with respect to model parameters?_
>
> > In Fig. 1, Fig. 4(a), Fig. 6, and Fig. 9, we have considered three different LLM as judges by adjusting their size from 70.6 B to 26.7 B and 17.9 B. Furthermore, as described in the reply  above, in the new experiment on test-time scaling for GSM8K data set, we have investigated the crucial impact of the choice of LLM as judges.
>
> 4) _Have you looked at scaling properties over risk with respect to the amount of human and synthetic data that is used?_
>
> > In Fig. 4(a) and Fig. 7(a), we have varied the amount of human data from 100 to 300 ($n: 100 \rightarrow 120 \rightarrow \cdots \rightarrow  300$) and observed its scaling trend. To further understand the scaling trend with the synthetic data size, we have carried out additional experiments that vary the size of synthetic data from 1000 to 6000 (\$N: 1000 \rightarrow 2000 \rightarrow \cdots \rightarrow 6000$) for a fixed amount of human data for the new math experiment. We adopt GPT-4.1 as the autoevaluator and the results are summarized in the following table. It is observed that all the schemes ensure the target risk (accuracy gain no less than 3%), and that autoeval-based approaches can increasingly save the reasoning budget with increased number of synthetic data. Interestingly, R-AutoEval+ shows a more robust behavior than R-AutoEval, demonstrating again the power of the proposed adaptive design.
>
> |  | R-Eval | R-AutoEval | R-AutoEval+ |
> | --- | --- | --- | --- |
> | test-time computation (num. tokens) | 983.34 | 995.84→917.33→899.53→883.99→867.36→858.94 | 889.53→866.89→865.96→856.13→852.47→849.31 |
> | accuracy gain | 7.00% | 6.98% →6.55% →6.43%→6.25%→6.16%→6.05% | 6.38% →6.22%→6.19%→6.11%→6.03%→6.03 % |
>
> **Table 3: Impact of the number of synthetic data \$N: 1000 \rightarrow 2000 \rightarrow3000 \rightarrow4000\rightarrow5000\rightarrow6000$**
>
> 5) _Would it be possible to evaluate this framework in a larger scale setting as the ones described above?_
>
> > In the new experiments on the GSM8K data set, we have considered a wide range of autoevaluators from model size from 0.85 B to 70.6 B, as well as for the GPT4.1 models which are generally assumed to be in a much larger scale setting. The results in Table 2 confirm the general applicability of the proposed scheme.
>
> #### **References**
> #### [Zheng et al., 2023] Lianmin Zheng, Wei-Lin Chiang, Ying Sheng, Siyuan Zhuang, Zhanghao Wu, Yonghao Zhuang, Zi Lin, Zhuohan Li, Dacheng Li, Eric Xing, et al. Judging llm-as-a-judge with mt-bench and chatbot arena. Advances in neural information processing systems, 36:46595–46623, 2023.
> #### [Li et al., 2025] Dawei Li, Renliang Sun, Yue Huang, Ming Zhong, Bohan Jiang, Jiawei Han, Xiangliang Zhang, Wei Wang, and Huan Liu. Preference leakage: A contamination problem in llm-as-a-judge. arXiv preprint arXiv:2502.01534, 2025.

---

### Official Review · Reviewer_nziA · 2025-07-03

**Clarity:** 3
**Significance:** 3
**Originality:** 4
**Rating:** 5
**Confidence:** 4

**Summary:**

This manuscript introduces "R-AutoEval+", an enhanced prediction-powered AutoEval method that builds upon previous R-AutoEval and Prediction-Powered Inference (PPI) work. R-AutoEval+ incorporates a dynamic parameter on top of PPI, which adjusts the weighting between human and automated evaluations. This adjustment is achieved through a "testing-by-betting" strategy, where weights are modified based on observations. This adaptation addresses R-AutoEval's limitations when the AutoRater's accuracy is insufficient.

The manuscript illustrates the method's application with two real-data examples:
(1) Selecting the smallest quantized LLMs with a guaranteed performance drop of less than 10%.
(2) Selecting the shortest prompt template with a guaranteed execution error of less than 0.5. In both cases, R-AutoEval+ demonstrated superior performance compared to R-Eval (pure human evaluation) and R-AutoEval (PPI-based AutoEval).

**Questions:**

- Since the weights are learned dynamically by the observations up to $i$, does the order of the data to the algorithm affect the results? How robust is the algorithm against the data ordering?

- Regarding the real-data experiments, how was the unlabeled data constructed for TriviaQA? How similar does the synthetic data need to be with the human labeled data? How were the hyperparameters, such as $n$, $N$, $S$, initial weight, synthetic-to-real, etc., determined?

- Similarly, for the instruction-induction task, how was the unlabeled data constructed? How many human evaluation and synthetic samples were used?

**Ethical Concerns:**

["NO or VERY MINOR ethics concerns only"]

**Final Justification:**

With the new results, I have a better understanding of the paper. I believe this is a solid work with good impact. I increases my confidence from 3 to 4.

Please add the new results and the relevant details to the revised paper. Thank you.

**Limitations:**

Yes

**Quality:**

4

**Strengths And Weaknesses:**

Strengths:
- The manuscript is very well written. The methodology and results are clearly presented. Even though the content is quite dense and deep, the authors were able to present it well with preliminaries, making it self-contained and easier for the reader to understand.

- The two real-data examples are cleverly chosen to demonstrate the method's use cases.

- Solid mathematical justification.

Weakness:
- The details of real data experiments could have been presented more thoroughly.

---

> ### Author Rebuttal · Authors · 2025-07-31
>
> Thank you for the constructive comments. Below, we provide a point-by-point response to each of the reviewer's comments.
>
> 1. _The details of real data experiments could have been presented more thoroughly._
>
> > If this work is accepted, we will make sure to add all the relevant details for the real-data experiments, including the method used for constructing the unlabeled data set and the measure of similarity between the human labeled data and synthetic data.
>
> 2. _Since the weights are learned dynamically by the observations up to , does the order of the data to the algorithm affect the results? How robust is the algorithm against the data ordering?_
>
> > Yes, the order of the data can affect the final result. To clarify this point, we have carried out additional experiments. Specifically, in light of the growing attention on test-time scaling of LLMs [Snell et al., 2024, Muennighoff et al., 2025], we applied our framework to the problem of identifying the minimum amount of additional computation that ensures desired performance gain. We adopted Qwen3-1.7 B as the base model as it provides both non-reasoning and reasoning modes [Yang et al., 2025]; and we considered the task of solving math problems (GSM8K) [Cobbe et al., 2021]. We allow the computation budget for the reasoning mode to vary between 128 to 1280 tokens, and our goal is to find the minimum reasoning budget that achieves at least extra 3% accuracy gain as compared to the non-reasoning mode. The table below shows the amount of test-time computation in the unit of tokens (i.e., the number of generated tokens to answer the math question) that can be saved by R-AutoEval+ as compared to R-Eval (4th column) and R-AutoEval (5th column) for autoevaluators with different sizes, accuracies on GSM8K. We set \$n=1000, N=4000, \delta=0.1$, and report the averaged output from 100 independent runs with different calibration-test data splits. We omit the accuracy gain results as all the schemes always achieve at least the target accuracy gain 3%. The table confirms again the efficiency and validity of the proposed framework.
>
>
> | autoevaluator spec (model name) | autoevaluator spec (model size) | autoevalator spec (acc) | test-time computation gain of R-AutoEval+ over R-Eval (num. tokens)  | test-time computation gain of R-AutoEval+ over R-AutoEval (num. tokens) |
> | :---: | :---: | :---: | :---: | :---: |
> | BitNet b1.58 | 0.85 B | 0.35 | 40.87 | 7.80 |
> | Llama-3.2-3B-Instruct  | 3.21 B | 0.66 | 90.48 | 7.72 |
> | DeepSeek-R1-Distill-Qwen-7B | 7.62 B | 0.54 | 56.49 | 14.81 |
>  DeepSeek-R1-0528-Qwen3-8B | 8.19 B | 0.32 | 21.40 | 24.59 |
> | Qwen3-32B | 32.8 B | 0.82 | 42.14 | 66.25 |
> | DeepSeek-R1-Distill-Qwen-32B | 32.8 B | 0.89 | 117.12 | 27.17 |
> | GPT-4.1 | not available | 0.9258 | 127.21 | 27.86 |
>
> **Table 1: LLM test-time scaling experiment for math problem (GSM8K)**
>
> >  To focus on the robustness of the algorithm to the ordering of the data, we fix the calibration-test data split and only change the ordering of data to compute the normalized deviation (std / mean) of  the test-time computation required by each algorithm. We will consider GPT-4.1 and DeepSeek-R1-0528-Qwen3-8B as the autoevaluators for the experimental results henceforth. As can be seen from the table below, R-AutoEval and R-AutoEval+ are more robust to the data ordering than R-Eval, possibly attributed to its reduced variability of averaged autoevaluated data (at each round we average \$r=N/n$ autoevaluated data) while R-AutoEval+ tends to be slightly less robust than R-AutoEval for its adaptive nature that incorporates the data ordering.
>
> |  | R-Eval | R-AutoEval | R-AutoEval+ |
> | --- | --- | --- | --- |
> | normalized deviation (std/mean) with autoeval = GPT4.1 | 8.62% | 3.90%| 6.88% |
> | normalized deviation (std/mean) with autoeval = DeepSeek-R1-0528-Qwen3-8B | 8.62% | 6.32%| 6.00% |
>
> **Table 2: Sensitivity of algorithms with respect to data ordering**
>
> 3. _Regarding the real-data experiments, how was the unlabeled data constructed for TriviaQA? How similar does the synthetic data need to be with the human labeled data? How were the hyperparameters, such as $n, N, S$, initial weight, synthetic-to-real, etc., determined?_
>
> > As TriviaQA does not explicitly contain unlabeled data, we have split the data set and ignored the presence of label information in one split to construct the unlabeled data set. The synthetic data indeed need to be similar enough with the human labeled data to ensure improved efficiency of R-AutoEval+. To clarify this similarity, we have measured the accuracy of the autoevaluator in the first table of this reply, which essentially captures the similarity of autoevaluated data with human labeled data.  The hyperparameter \$n$ was searched within the interval 100 to 300 as shown in Fig. 4(a), while we have chosen the hyperparameter $N$ based on the amount available unlabeled data (from the above construction).  In our experiments, we have chosen the hyperparameters associated with the candidate factors by setting $S=10$ with equal grid and equal weighting, since such setting gave us a good result in toy experiment. Nonetheless, we have further carried out careful experimental analysis on the impact of hyperparameter choice during the rebuttal.
>
> > We first investigate the impact on the number of candidates, S, with candidate factors equally spaced within the interval [0,1]; and with equal initial weights.  In what follows, we will adopt GPT-4.1 and DeepSeek-R1-0528-Qwen3-8B as the autoevaluators.
>
> |  | R-Eval | R-AutoEval | R-AutoEval+(S=2) | R-AutoEval+(S=5) | R-AutoEval+(S=10) | R-AutoEval+(S=20) | R-AutoEval+(S=50) |
> | --- | --- | --- | --- | --- | --- | --- | --- |
> | test-time computation in tokens (autoeval = GPT4.1) | 983.34 | 883.99 | 871.67 | 854.99 | 856.13 | 856.13 | 856.13 |
> | test-time computation  in tokens (autoeval = DeepSeek-R1-0528-Qwen3-8B) | 983.34 | 986.53 | 964.41 | 961.94 | 961.94 | 961.94 | 961.94 |
>
> **Table 2: Impact of the number of candidate factors (\$S$)**
>
> > As can be observed in the table above, as long as the candidate factors includes \$\rho=0$ (R-Eval) and \$\rho=1$ (R-AutoEval), i.e., as long as \$S \geq 2$, R-AutoEval+ consistently outperforms both R-Eval and R-AutoEval, with its gain being maximized at $S=5$, and saturates after $S=10$.
>
> > Next, we investigate the impact of the values of each candidate factors by going beyond the equally spaced grid. To this end, we consider Dirichlet distribution with concentration parameter 0.1, 1, 10 to allocate the $S-2$ values within the [0,1] interval (we keep the first and the last factors to be 0 and 1 to ensure the efficiency guarantee as per our Theorem 3). The cumulative probabilities of the Dirichlet distribution are used as the candidate factors. An increased concentration parameter makes samples from the Dirichlet distribution closer to the uniform distribution, and hence the grid becomes closer to the equally spaced grid. The table below shows that equal grid performs the best.
>
> |  | R-Eval | R-AutoEval | R-AutoEval+ (equal grid) | R-AutoEval+  (Dir(10,10,...,10)) | R-AutoEval+  (Dir(1,1,...,1)) | R-AutoEval+  (Dir(0.1,0.1,...,0.1)) |
> | --- | --- | --- | --- | --- | --- | --- |
> |  test-time computation in tokens (autoeval = GPT4.1)  | 983.34 | 883.99 | 856.13 | 856.13 | 860.91 | 873.27 |
> |  test-time computation in tokens (autoeval = DeepSeek-R1-0528-Qwen3-8B) | 983.34 | 986.53 | 961.94 | 964.22 | 960.76 | 966.72 |
>
> **Table 3: Impact of the value of candidate factors (\$\\{\rho_s\\}_{s=1}^S$)**
>
> > The last remaining degree of freedom is given by the choice of the weights associated with each candidate factor. To this end, we go beyond the uniform distribution and consider again the Dirichlet distribution to allocate the weights for the \$S$ candidate factors. The table below confirms again the preference for equal weighting.
>
> |  | R-Eval | R-AutoEval | R-AutoEval+ (equal weight) | R-AutoEval+  (Dir(10,10,...,10)) | R-AutoEval+  (Dir(1,1,...,1)) | R-AutoEval+  (Dir(0.1,0.1,...,0.1)) |
> | --- | --- | --- | --- | --- | --- | --- |
> |  test-time computation in tokens (autoeval = GPT4.1) | 983.34 | 883.99 | 856.13 | 856.13 | 856.13 | 856.68 |
> |  test-time computation in tokens (autoeval = DeepSeek-R1-0528-Qwen3-8B) | 983.34 | 986.53 | 961.94 | 959.21 | 962.41 | 969.50 |
>
> **Table 4: Impact of the weights associated with the candidate factors (\$\\{w_{s,0}\\}_{s=1}^S$)**
>
>
> 4. _Similarly, for the instruction-induction task, how was the unlabeled data constructed? How many human evaluation and synthetic samples were used?_
>
> > We followed the same approach as TriviaQA, and we have considered 200 human evaluation and 1800 synthetic samples.
>
> #### **References**
> #### [Cobbe et al., 2021] Karl Cobbe, et al. Training verifiers to solve math word problems. arXiv preprint arXiv:2110.14168, 2021.
> #### [Muennighoff et al., 2025] Niklas Muennighoff,et al. s1: Simple test-time scaling. arXiv preprint arXiv:2501.19393, 2025.
> #### [Snell et al., 2024] Charlie Snell, et al. Scaling llm test-time compute optimally can be more effective than scaling model parameters. arXiv preprint arXiv:2408.03314, 2024.

---

> > ### Comment · Reviewer_nziA · 2025-08-04
> >
> > Thank you for the new experimental results and ablations. They are very helpful for me to better understand the proposed method.
> >
> > Do you have insights on why in Table 1, GPT-4.1 has a higher autoevalator accuracy (0.9258) than Qwen3-32B (0.82), but its test-time computation gain of R-AutoEval+ over R-AutoEval is lower (27.86 vs 66.25). What are the other factors that affect the results? Thank you!

---

> > > ### Author Response · Authors · 2025-08-04
> > >
> > > Thank you again for the constructive comments! We will make sure to add all the new experimental results and ablations in the final version if this work gets accepted.
> > >
> > > Your point about Table 1 refers to an interesting behavior that stems from the positive feedback of LLM judges within the same family [Zheng et al., 2023], also known as preference leakage [Li et al., 2025]. Preference leakage may cause an accurate autoevaluator to degrade the precision of a risk estimate, as you observed for Qwen3-32B. This degradation can be explained by the correlation between the outputs of the autoevaluator and of the model being evaluated. This correlation can entail a form of positive feedback between model and autoevaluator, making it more difficult to debias the empirical estimates provided by the autoevaluator.
> > >
> > > To make this point clear, we have updated Table 1 for the mentioned two autoevaluators by explicitly showing the test-time computation of R-Eval, R-AutoEval, and R-AutoEval+ in the table below. As seen from the table, when employing Qwen3-32B (which is in the same family of the base reasoning Qwen3-1.7B model) as the autoevaluator, R-AutoEval shows worse test-time computation (1007.45) than R-Eval (983.34). Please note that, even under such preference leakage, R-AutoEval+ achieves better test-time computation (941.2) than R-Eval (983.34).
> > >
> > > | autoevaluator spec (model name) | **autoevaluator spec (model size)** | **autoevalator spec (acc)** | test-time computation of R-Eval (num. tokens) | test-time computation of R-AutoEval (num. tokens) | test-time computation of R-AutoEval+ (num. tokens) |
> > > | --- | --- | --- | --- | --- | --- |
> > > | Qwen3-32B | 32.8 B | 0.82 | 983.34 | 1007.45 | 941.2 |
> > > | GPT-4.1 | not available | 0.9258 | 983.34 | 883.99 | 856.13 |
> > >
> > > **Table 1’:  LLM test-time scaling experiment for math problem (GSM8K) with absolute test-time computation for each scheme**
> > >
> > >
> > > **References**
> > >
> > > [Zheng et al., 2023] Lianmin Zheng, Wei-Lin Chiang, Ying Sheng, Siyuan Zhuang, Zhanghao Wu, Yonghao Zhuang, Zi Lin, Zhuohan Li, Dacheng Li, Eric Xing, et al. Judging llm-as-a-judge with mt-bench and chatbot arena. Advances in neural information processing systems, 36:46595–46623, 2023.
> > >
> > > [Li et al., 2025] Dawei Li, Renliang Sun, Yue Huang, Ming Zhong, Bohan Jiang, Jiawei Han, Xiangliang Zhang, Wei Wang, and Huan Liu. Preference leakage: A contamination problem in llm-as-a-judge. arXiv preprint arXiv:2502.01534, 2025.

---

> > > > ### Comment · Reviewer_nziA · 2025-08-05
> > > >
> > > > Thank you very much for the explanation and the new results. That makes sense to me. Really appreciate it!

---

### Official Review · Reviewer_3kvC · 2025-07-22

**Clarity:** 2
**Significance:** 2
**Originality:** 2
**Rating:** 4
**Confidence:** 2

**Summary:**

This paper introduces R-AutoEval+, a novel framework for evaluating AI models, particularly LLMs. It addresses the challenge of balancing between costly real-world data evaluations and potentially biased automated evaluations. R-AutoEval+ provides finite-sample reliability guarantees while ensuring improved or at least equivalent sample efficiency compared to conventional methods. The key innovation is an adaptive construction of the model evaluation variable that dynamically adjusts its reliance on synthetic data based on the autoevaluator's accuracy. The framework leverages game-theoretic testing-by-betting and an enhanced version of prediction-powered inference (PPI++). The authors demonstrate the reliability and efficiency of R-AutoEval+ through experiments on LLM weight quantization optimization and prompt design.

**Questions:**

- What's the computational overhead of the algorithm compared to other methods?
- How sensitive is the performance on the selection of hyperparameters?

The paper goes into mathematical details quite quickly. It would help if a more high-level problem statement was given in the introduction, before going more into depth.

**Ethical Concerns:**

["NO or VERY MINOR ethics concerns only"]

**Final Justification:**

The authors response has clarified the main points and provided the requested evidence. I maintain my score of 4.

**Limitations:**

Yes

**Paper Formatting Concerns:**

No issues identified

**Quality:**

3

**Strengths And Weaknesses:**

Strengths:
- Provides finite-sample reliability guarantees
- Adaptively balances the use of synthetic and real-world data, potentially improving sample efficiency
- Offers provable sample efficiency improvements over existing methods
- Combines multiple advanced techniques (game-theoretic testing-by-betting, PPI++)
- Demonstrates practical applicability through experiments on relevant LLM tasks
- Addresses a crucial problem in AI model evaluation and selection

Weaknesses:
- Uses a fixed, discrete set of candidate factors for determining reliance on synthetic data, potentially limiting flexibility
- Sample efficiency guarantee only holds for sufficiently high target reliability levels
- Limited experimental validation (only tested on LLM quantization and prompt design)

---

> ### Author Rebuttal · Authors · 2025-07-31
>
> Thank you for the constructive comments. Below, we provide a point-by-point response to each of the reviewer's comments.
>
> 1) _Limited experimental validation (only tested on LLM quantization and prompt design)_
>
> > During the rebuttal, we have carried out additional experiments on more challenging LLM tasks. In light of the growing attention on test-time scaling of LLMs [Snell et al., 2024, Muennighoff et al., 2025], we applied our framework to the problem of identifying the minimum amount of additional computation that ensures desired performance gain. We adopted Qwen3-1.7 B as the base model as it provides both non-reasoning and reasoning modes [Yang et al., 2025]; and we considered the task of solving math problems (GSM8K) [Cobbe et al., 2021]. We allow the computation budget for the reasoning mode to vary between 128 to 1280 number of tokens, and our goal is to find the minimum reasoning budget that achieves at least extra 3% accuracy gain as compared to the non-reasoning mode. The table below shows the amount of test-time computation in the unit of tokens (i.e., the number of generated tokens to answer the math question) that can be saved by R-AutoEval+ as compared to R-Eval (4th column) and R-AutoEval (5th column) for autoevaluators with different sizes, accuracies on GSM8K. We set \$n=1000, N=4000, \delta=0.1$, and report the averaged output from 100 independent runs. We omit the accuracy gain results as all the schemes always achieve at least the target accuracy gain 3%. The table confirms again the efficiency and validity of the proposed framework.
>
> | autoevaluator spec (model name) | autoevaluator spec (model size) | autoevalator spec (acc) | test-time computation gain of R-AutoEval+ over R-Eval (num. tokens)  | test-time computation gain of R-AutoEval+ over R-AutoEval (num. tokens) |
> | :---: | :---: | :---: | :---: | :---: |
> | BitNet b1.58 | 0.85 B | 0.35 | 40.87 | 7.80 |
> | Llama-3.2-3B-Instruct  | 3.21 B | 0.66 | 90.48 | 7.72 |
> | Qwen3-32B | 32.8 B | 0.82 | 42.14 | 66.25 |
> | DeepSeek-R1-Distill-Qwen-32B | 32.8 B | 0.89 | 117.12 | 27.17 |
> | GPT-4.1 | not available | 0.9258 | 127.21 | 27.86 |
>
> **Table 1: LLM test-time scaling experiment for math problem (GSM8K)**
>
> > In the following, we will use the new setting to address the reviewer’s comments.
>
> 2) _Uses a fixed, discrete set [...], potentially limiting flexibility_
>
> > This is indeed a limitation of the scheme as we briefly described in Sec. 5 (Conclusion and Further Discussions). To further explore the impact of the selection of candidate factors, we have carried out an experiment in which we vary the number of candidates, S, which are equally spaced within the interval [0,1]. In what follows, we will adopt GPT-4.1 and DeepSeek-R1-0528-Qwen3-8B as the autoevaluators.
>
> |  | R-Eval | R-AutoEval | R-AutoEval+(S=2) | R-AutoEval+(S=5) | R-AutoEval+(S=10) | R-AutoEval+(S=20) | R-AutoEval+(S=50) |
> | --- | --- | --- | --- | --- | --- | --- | --- |
> | test-time computation in tokens (autoeval = GPT4.1) | 983.34 | 883.99 | 871.67 | 854.99 | 856.13 | 856.13 | 856.13 |
> | test-time computation  in tokens (autoeval = DeepSeek-R1-0528-Qwen3-8B) | 983.34 | 986.53 | 964.41 | 961.94 | 961.94 | 961.94 | 961.94 |
>
> **Table 2: Impact of the number of candidate factors (\$S$)**
>
> > As can be observed in the table above, as long as the candidate factors includes \$\rho=0$ (R-Eval) and \$\rho=1$ (R-AutoEval), i.e., as long as \$S \geq 2$, R-AutoEval+ consistently outperforms both R-Eval and R-AutoEval, with its gain being maximized at $S=5$, and saturates after $S=10$.
>
> > Next, we investigate the impact of the values of each candidate factors by going beyond the equally spaced grid. To this end, we consider Dirichlet distribution with concentration parameter 0.1, 1, 10 to allocate the $S-2$ values within the [0,1] interval (we keep the first and the last factors to be 0 and 1 to ensure the efficiency guarantee as per our Theorem 3). The cumulative probabilities of the Dirichlet distribution are used as the candidate factors. An increased concentration parameter makes samples from the Dirichlet distribution closer to the uniform distribution, and hence the grid becomes closer to the equally spaced grid. The table below shows that equal grid performs the best.
>
> |  | R-Eval | R-AutoEval | R-AutoEval+ (equal grid) | R-AutoEval+  (Dir(10,10,...,10)) | R-AutoEval+  (Dir(1,1,...,1)) | R-AutoEval+  (Dir(0.1,0.1,...,0.1)) |
> | --- | --- | --- | --- | --- | --- | --- |
> |  test-time computation in tokens (autoeval = GPT4.1)  | 983.34 | 883.99 | 856.13 | 856.13 | 860.91 | 873.27 |
> |  test-time computation in tokens (autoeval = DeepSeek-R1-0528-Qwen3-8B) | 983.34 | 986.53 | 961.94 | 964.22 | 960.76 | 966.72 |
>
> **Table 3: Impact of the value of candidate factors (\$\\{\rho_s\\}_{s=1}^S$)**
>
> > The last remaining degree of freedom is given by the choice of the initial weights associated with each candidate factor. To this end, we go beyond the uniform distribution and consider again the Dirichlet distribution to allocate the weights for the \$S$ candidate factors. The table below confirms again the preference for equal weighting.
>
> |  | R-Eval | R-AutoEval | R-AutoEval+ (equal weight) | R-AutoEval+  (Dir(10,10,...,10)) | R-AutoEval+  (Dir(1,1,...,1)) | R-AutoEval+  (Dir(0.1,0.1,...,0.1)) |
> | --- | --- | --- | --- | --- | --- | --- |
> |  test-time computation in tokens (autoeval = GPT4.1) | 983.34 | 883.99 | 856.13 | 856.13 | 856.13 | 856.68 |
> |  test-time computation in tokens (autoeval = DeepSeek-R1-0528-Qwen3-8B) | 983.34 | 986.53 | 961.94 | 959.21 | 962.41 | 969.50 |
>
> **Table 4: Impact of the initial weights associated with the candidate factors (\$\\{w_{s,0}\\}_{s=1}^S$)**
>
> 3) _Sample efficiency [...] high target reliability levels_
>
> > This is indeed a limitation of the study, as described in Sec. 5 (Conclusion and Further Discussions). That said, in practice, autoeval methods may be particularly relevant for safety-critical risk-averse scenarios [Giambona et al., 2018], in which high reliability levels are a requirement. Furthermore, we have observed empirically that, even under moderately low reliability levels \$1-\delta$ (with larger \$\delta$ indicating a less strict reliability requirement), R-AutoEval+ consistently outperforms R-Eval and R-AutoEval. For the ongoing example, this is shown in the following table.
>
> |  | \$\delta = 0.05$ | \$\delta = 0.1$  | \$\delta = 0.15$  | \$\delta = 0.2$  | \$\delta = 0.25$  | \$\delta = 0.3$  |
> | --- | --- | --- | --- | --- | --- | --- |
> | test-time computation gain over R-Eval | 133.72 | 127.21 | 110.34 | 102.88 | 99.29 | 87.65 |
> | test-time computation gain over R-AutoEval | 34.20 | 27.86 | 14.41 | 4.40 | 1.24 | 1.47 |
>
> **Table 5: Efficiency gain for a range of target reliability levels \$\delta$**
>
> 4) _What's the computational overhead of the algorithm compared to other methods?_
>
> > Given both the human-labeled (size \$n$) and autoevaluated data (size \$N$), mean-based approaches (Eval and AutoEval, which do not come with reliability guarantees) require \$O(n)$ (for Eval) and \$O(N)$ (for AutoEval) operations for computing the respective average. The reliable approaches (R-Eval and R-AutoEval) need additional computation steps for updating the e-values as well as the betting strategies (Sec. 2). While updating the e-values at each round requires a single multiplication, the betting strategy may be more complex. For example, especially the universal portfolio (UP) strategy with sublinear regret (Assumption A1) has a complexity of order \$O(nG)$ where $G$ is the size of the grid used to approximate the integral operation of UP (Sec. C.1), although this can be decreased by adopting sampling-based approaches [Kalai and Vempala, 2002]. Lastly, R-AutoEval+ requires $S$ times more computation than R-AutoEval due to its consideration of $S$ candidate factors. It is worth emphasizing that, the costs described above are generally negligible with respect to running the autoevaluator (e.g., an LLM judge). The summary of the computation complexities can be found in the below table.
>
> |  | Eval | AutoEval | R-Eval | R-AutoEval | R-AutoEval+ |
> | --- | --- | --- | --- | --- | --- |
> | computation complexity | \$O(n)$ | \$O(N)$ | \$O(n + nG)$ | \$O(n r + nG)$ | \$O(S n r + SnG)$ |
>
> **Table 6: Computational complexities of the algorithms considered in this work**
>
> 5) _How sensitive [...] hyperparameters?_
>
> > The hyperparameters in the proposed R-AutoEval+ encompass the number of candidate factors, the values of the candidate factors,  and the weights associated with the candidate factors, whose sensitivity has been analyzed in the reply  above.
>
> 6) _The paper goes into mathematical details quite [...] depth._
>
> >  We have tried to provide a high-level discussion through the introduction section and Fig. 1, but the page limitations at the time of submission made it difficult to provide further general descriptions. We will certainly add further discussions on the high-level problem statement using the additional content page in case the work is accepted.
>
> #### **References**
> #### [Cobbe et al., 2021] Karl Cobbe, et al. Training verifiers to solve math word problems. arXiv preprint arXiv:2110.14168, 2021.
>
> #### [Muennighoff et al., 2025] Niklas Muennighoff,et al. s1: Simple test-time scaling. arXiv preprint arXiv:2501.19393, 2025.
>
> #### [Snell et al., 2024] Charlie Snell, et al. Scaling llm test-time compute optimally can be more effective than scaling model parameters. arXiv preprint arXiv:2408.03314, 2024.
>
> #### [Yang et al., 2025] An Yang, et al. Qwen3 technical report. arXiv preprint arXiv:2505.09388, 2025.
>
> #### [Giambona et al., 2018] Erasmo Giambona, et al. The theory and practice of corporate risk management: Evidence from the field. Financial Management, 47(4):783–832, 2018.
>
> #### [Kalai and Vempala, 2002] Adam Kalai and Santosh Vempala. Efficient algorithms for universal portfolios. Journal of Machine Learning Research, 3(Nov):423–440, 2002.

---

> ### Comment · Reviewer_3kvC · 2025-08-07
>
> Thank you for providing additional details and clarifications in the rebuttal.

---

### Decision · Program_Chairs · 2025-09-17

**Decision:**

Accept (spotlight)

**Comment:**

1. **Summary**
   The paper introduces R-AutoEval+, a new framework for autoevaluation that adaptively combines real human-labeled data and synthetic evaluations from LLMs-as-judges. The central contribution is a method that ensures finite-sample reliability guarantees while providing sample efficiency improvements (or at least never worse performance) compared to both R-Eval and R-AutoEval. The method builds on testing-by-betting with e-values and an extension of prediction-powered inference (PPI++), adaptively weighting reliance on autoevaluators. Empirical demonstrations include (i) selecting quantized LLMs under accuracy guarantees, (ii) prompt selection with execution risk guarantees, and (iii) rebuttal-period experiments on test-time compute scaling for reasoning LLMs.

2. **Strengths**

   1. Strong theoretical foundation, with rigorous proofs of finite-sample guarantees and efficiency results.
   2. Novel adaptive design that seamlessly reverts to human-only evaluation when autoevaluators are poor, mitigating known risks of prior methods.
   3. Solid empirical validation across several tasks (LLM quantization, prompt design, test-time scaling), demonstrating both reliability and efficiency.
   4. Clear exposition of the methodological innovations (testing-by-betting, adaptive weighting, PPI++).
   5. Active engagement by the authors during rebuttal, with new experiments and ablations that addressed reviewer concerns.

3. **Weaknesses**

   1. Experimental coverage, while improved during rebuttal, remains narrower than ideal; the tasks are limited to LLM quantization, prompting, and one test-time scaling example. A broader set of application domains would better substantiate generality.
   2. Efficiency guarantees hold only under sufficiently high reliability levels. Although empirically the method still performs well at moderate levels, the theoretical coverage is narrower.
   3. Dependence on a fixed discrete set of candidate factors for weighting synthetic data may limit flexibility, though rebuttal experiments suggest performance is stable under different configurations.
   4. Computational overhead, while negligible compared to running LLM evaluators, is higher than for R-AutoEval due to the adaptive weighting.

4. **Decision Rationale**

   * The paper makes a clear and rigorous theoretical advance, addressing a critical limitation of prior autoevaluation frameworks (the trade-off between bias correction and efficiency).
   * The empirical evidence, although somewhat narrow in scope, convincingly demonstrates the utility of the method and its robustness across evaluators and tasks.
   * The rebuttal period was particularly constructive: the authors added experiments on test-time scaling, evaluated across multiple model families, and explored robustness to hyperparameter choices, addressing key reviewer concerns.
   * Despite some remaining limitations, the work is technically solid, timely, and relevant for the community given the centrality of evaluation in current LLM research and deserve wider dissemination.

   For these reasons, I recommend **acceptance at the spotlight level**.

5. **Discussion and Reviewer Exchanges**

   * **Reviewer 3kvC**: Raised concerns about limited experiments, discrete factors, and high-level clarity. Authors responded with additional experiments on test-time scaling and ablation studies on factor discretization and weights. Reviewer maintained a borderline accept score.
   * **Reviewer nziA**: Initially positive, with questions on robustness to data ordering and experiment details. Authors clarified experimental construction, robustness properties, and provided new sensitivity results. Reviewer increased confidence and remained supportive of acceptance.
   * **Reviewer UVVV**: Expressed concerns about toyish experiments and limited evaluator families. Authors responded with extended experiments on multiple autoevaluators across families and scaling of synthetic data. Reviewer maintained a borderline accept with recommendation to accept.
   * **Reviewer dCNW**: Appreciated the clarity and theory but questioned significance and practical impact. Authors responded with additional significance experiments and clarifications. Reviewer maintained borderline accept.

   Overall, the discussion improved the paper significantly. While some reviewers remained conservative due to limited experimental scope, the theoretical contributions and new rebuttal results consolidated the consensus beyond acceptance.